# Calcium handling precedes cardiac differentiation to initiate the first heartbeat

Richard CV Tyser[1,2†], Antonio MA Miranda[1†], Chiann-mun Chen[3], Sean M Davidson[2], Shankar Srinivas[1*‡], Paul R Riley[1*‡]

[1]Department of Physiology, Anatomy and Genetics, University of Oxford, Oxford, United Kingdom; [2]The Hatter Cardiovascular Institute, University College London and Medical School, London, United Kingdom; [3]Wellcome Trust, London, United Kingdom

*For correspondence: shankar. srinivas@dpag.ox.ac.uk (SS); paul. riley@dpag.ox.ac.uk (PRR)

[†]These authors contributed equally to this work
[‡]These authors also contributed equally to this work

Competing interests: The authors declare that no competing interests exist.

**Abstract** The mammalian heartbeat is thought to begin just prior to the linear heart tube stage of development. How the initial contractions are established and the downstream consequences of the earliest contractile function on cardiac differentiation and morphogenesis have not been described. Using high-resolution live imaging of mouse embryos, we observed randomly distributed spontaneous asynchronous $Ca^{2+}$-oscillations (SACOs) in the forming cardiac crescent (stage E7.75) prior to overt beating. Nascent contraction initiated at around E8.0 and was associated with sarcomeric assembly and rapid $Ca^{2+}$ transients, underpinned by sequential expression of the $Na^+$-$Ca^{2+}$ exchanger (NCX1) and L-type $Ca^{2+}$ channel (LTCC). Pharmacological inhibition of NCX1 and LTCC revealed rapid development of $Ca^{2+}$ handling in the early heart and an essential early role for NCX1 in establishing SACOs through to the initiation of beating. NCX1 blockade impacted on CaMKII signalling to down-regulate cardiac gene expression, leading to impaired differentiation and failed crescent maturation.

## Introduction

The heart is the first organ to form and function during mammalian embryonic development. In the mouse, mesoderm originating from the primitive streak forms a bilateral pool of progenitor cells that at E7.5 give rise to the cardiac crescent (CC). The CC subsequently expands and migrates to the midline whereupon, between E8.25 and E8.5, the two sides of the CC fuse and form the linear heart tube (LHT) (reviewed in *Buckingham et al., 2005*). The first cardiac contractions have been described during the transition from CC to LHT. Studies of heart development in model organisms have historically focused on the origin and spatial-temporal allocation of cardiac progenitors and cardiovascular lineage determination (*Buckingham et al., 2005*; *Saga et al., 1996*; *Cai et al., 2003*; *Meilhac et al., 2004*; *Wu et al., 2006*; *Moretti et al., 2006*; *Evans et al., 2010*; *Devine et al., 2014*). Whilst insight into the identification and regulation of cardiac cell types is important for improved understanding of congenital heart disease (*Bruneau, 2008*), an anatomical and cellular bias has overlooked a role for the onset of cardiac function. Early descriptions of initial cardiac contractions (*Navaratnam et al., 1986*), including suggested pacemaker activity on either side of the embryonic midline (*Goss, 1952*), as well as optical mapping of spontaneous action potentials performed in both chicken (7 somite stage) and rat (3-somite stage) (*Fujii et al., 1981*; *Hirota et al., 1985*), have been informative, but lack resolution. Subsequent studies in mouse (3-somite stage) could only infer early electrical activity based on irregular fluctuations in basal $Ca^{2+}$ (*Nishii and Shibata, 2006*). Given the forming heart contracts from an early stage, this raises the important

**eLife digest** The heart is the first organ to form and to begin working in an embryo during pregnancy. It must begin pumping early to supply oxygen and nutrients to the developing embryo. Coordinated contractions of specialised muscle cells in the heart, called cardiomyocytes, generate the force needed to pump blood. The flow of calcium ions into and out of the cardiomyocytes triggers these heartbeats. In addition to triggering heart contractions, calcium ions also act as a messenger that drives changes in which genes are active in the cardiomyocytes and how these cells behave.

Scientists commonly think of the first heartbeat as occurring after a tube-like structure forms in the embryo that will eventually develop into the heart. However, it is not yet clear how the first heartbeat starts or how the initial heartbeats affect further heart development.

Tyser, Miranda et al. now show that the first heartbeat actually occurs much earlier in embryonic development than widely appreciated. In the experiments, videos of live mouse embryos showed that prior to the first heartbeat the flow of calcium ions between different cardiomyocytes is not synchronised. However, as the heart grows these calcium flows become coordinated leading to the first heartbeat. The heartbeats also become faster as the heart grows. Using drugs to block the movement of calcium ions, Tyser, Miranda et al. also show that a protein called NCX1 is required to trigger the calcium flows prior to the first heartbeat. Moreover, the experiments revealed that these early heartbeats help drive the growth of cardiomyocytes and shape the developing heart.

Together, the experiments show that the first heartbeats are essential for normal heart development. Future studies are needed to determine what controls the speed of the first heartbeats, and what organises the calcium flows that trigger the first heartbeat. Such studies may help scientists better understand birth defects of the heart, and may suggest strategies to rebuild hearts that have been damaged by a heart attack or other injury.

question of when and how contractile activity of cardiomyocytes is first initiated during development and to what extent this influences the progression of differentiation and subsequent cardiac morphogenesis. This is especially important as the forces exerted by cardiac contractions have been shown in several models to be required for proper heart development (*Granados-Riveron and Brook, 2012*), and to modulate gene expression (*Miyasaka et al., 2011*) at later developmental stages.

In mature cardiomyocytes, coordinated electrical excitation is coupled to physical contraction in a process termed excitation contraction coupling (ECC) (*Bers, 2002*). ECC relies on changes in the intracellular concentration of the second messenger $Ca^{2+}$ via release from the sarcoplasmic reticulum (SR) in a process termed $Ca^{2+}$ induced $Ca^{2+}$ release (CICR) (*Fabiato and Fabiato, 1979*). Increases in the concentration of intracellular $Ca^{2+}$ result in cardiomyocyte contraction due to $Ca^{2+}$ binding to troponin and myofilament activation. ECC involves a number of specific proteins including L-type $Ca^{2+}$ channels (LTCC, sarcolemmal $Ca^{2+}$ influx), ryanodine receptors (RyR2, SR $Ca^{2+}$ release), the sarco (endo) plasmic reticulum $Ca^{2+}$ ATPase (SERCA, SR $Ca^{2+}$ uptake) and the $Na^+/Ca^{2+}$ exchanger (NCX, Sarcolemmal $Ca^{2+}$ efflux). Targeted disruption of genes encoding ECC proteins in mice has shown that contractile activity of immature cardiomyocytes does not require ECC. Embryonic cardiomyocytes have a less developed SR and T-tubule system as well as an increased requirement for sarcolemmal $Ca^{2+}$ flux (*Conway et al., 2002*; *Seki et al., 2003*) and whilst they express a variety of ion channels and exchangers present in the adult heart (*Seisenberger et al., 2000*; *Cribbs et al., 2001*; *Linask et al., 2001*), the expression and activity of these proteins is distinct from that in mature cardiomyocytes (*Liang et al., 2010*). Using isolated cells as well as genetically manipulated animals, two contrasting mechanisms have been proposed for how $Ca^{2+}$ transients are generated in the developing heart from approximately E8.5 onwards. Early studies suggested that myocyte contraction is triggered by sarcolemmal $Ca^{2+}$ influx through voltage activated $Ca^{2+}$ channels with little or no contribution from the SR (*Nakanishi et al., 1988*; *Takeshima et al., 1998*). In contrast, more recently it has been shown that at ~E8.5–9, $Ca^{2+}$ transients originate from the SR, via RyR together with $InsP_3$ channels, to trigger electrical activity as well as contraction (*Viatchenko-Karpinski et al., 1999*;

*Méry et al., 2005*; *Sasse et al., 2007*; *Rapila et al., 2008*). Whilst these studies characterised SR function at ~E8.5–9, they did not investigate how $Ca^{2+}$ transients are regulated at the earliest stages of cardiac crescent development when contraction is initiated, and relied on experiments performed using isolated cells cultured for between 12 to 70 hr (*Sasse et al., 2007*; *Rapila et al., 2008*). Thus there is a lack of cellular resolution *in vivo* and no current mechanistic insight into the onset of $Ca^{2+}$ handling and its impact on differentiation and cardiogenesis.

We report here, for the first time, high-resolution live imaging of $Ca^{2+}$ transients during the earliest manifestation of murine heart development well before any indication of spontaneous cardiac contractions. We employed the use of multiple pharmacological inhibitors to address the contribution of the NCX1 and LTCC $Ca^{2+}$ channels during this process and reveal an essential early role for NCX1-dependent $Ca^{2+}$ handling on downstream cardiac differentiation and morphogenesis.

## Results

### Staging of early cardiac development and sarcomeric assembly

It is commonly stated that initiation of contraction begins with the formation of the LHT (*Bruneau, 2008*), and whilst cardiac contractions have been reported just prior to the 'linear heart tube' stage (*Navaratnam et al., 1986*; *Nishii and Shibata, 2006*; *Linask et al., 2001*; *Kumai et al., 2000*; *Porter and Rivkees, 2001*), a precise study on the initiation of cardiac function has not been conducted. A difficulty with these reports is the use of 'embryonic day' or 'somite number' to stage the developing heart. Somite number is variable in its correlation to the overall embryonic stage (*Kaufman and Navaratnam, 1981*), can depend on genetic background (*Méry et al., 2005*; *Porter and Rivkees, 2001*) and importantly, is not a sufficiently fine-grained proxy for the developmental stages of the heart. This can lead to ambiguities, as a '3-somite' embryo may range from the cardiac crescent to early LHT stages. We, therefore, created a staging system specific to the early heart, from early crescent to LHT (*Supplementary file 1a*), similar to studies at later stages when a more precise morphological characterization is necessary (*Biben and Harvey, 1997*). On this basis, we defined four stages (0, 1, 2 and 3) of cardiac crescent development prior to the LHT stage, based on clear morphological differences. Stage 0 hearts represented the first discernible crescent structure situated beneath the developing head folds, being the widest (360–390 μm along the mediolateral axis) and thinnest (70–80 μm along the rostro-caudal axis) of the crescent stages (*Supplementary file 1a*). Whilst stage 1 was morphologically similar to stage 0, the cardiac crescent had become narrower (300–370 μm) and thicker (75–95 μm). By stage 2, folding of the cardiac crescent is evident based on the formation of a trough at the embryonic midline and two lobes on either side. As the embryo transitions to stage 3 this trough becomes less obvious with a rostral-caudal elongation of the heart as the LHT begins to form. Transition from stage 3 to the LHT was defined by the complete fusion of the two lobes and loss of the central trough (*Figure 1*; *Supplementary file 1a*).

To molecularly characterize these stages, we performed immunostaining for three proteins of the cardiac contractile machinery: sarcomeric α-actinin (α-Act), a protein of the Z-line; Myomesin (Myom), a protein of the M-line and cardiac Troponin T (cTnT), the Tropomyosin binding subunit of the troponin complex (*Figure 1*; *Figure 1—figure supplement 1*). At stage 0, cTnT was the most evident contractile protein within the early cardiac crescent, albeit without sarcomeric banding (*Figure 1—figure supplement 1A*). Both α-Actinin and Myomesin appeared in small clusters of cells at stage 0 (*Figure 1A*). Sarcomeric banding of these proteins, indicative of contractile capability, only manifested later in discrete regions within stage 1 and 2 crescents (*Figure 1B,C*; *Figure 1—figure supplement 1B,C*). This was surprising given that the crescent stages of early heart development are thought to correspond to non-differentiated cardiac mesoderm (reviewed in *Harvey, 2002*), without prior reports of contractile machinery or functionality. By stage 3, sarcomere assembly and myofibrilar banding became more uniform, coincident with coalescence of the paired crescent primordia to the embryonic midline. This increased through to the linear heat tube stage, consistent with progressive cardiomyocyte maturation (*Figure 1D,E*; *Figure 1—figure supplement 1D,E*). To further characterize stage development in relation to sarcomere assembly, qRT-PCR was performed on isolated cardiac crescents, to assess the corresponding gene expression of *Myom1* (encoding Myom), *Actn2* (encoding α-actinin) and *Tnnt2* (encoding cTnT) (*Figure 1F*). *Myom1*, *Actn2* and

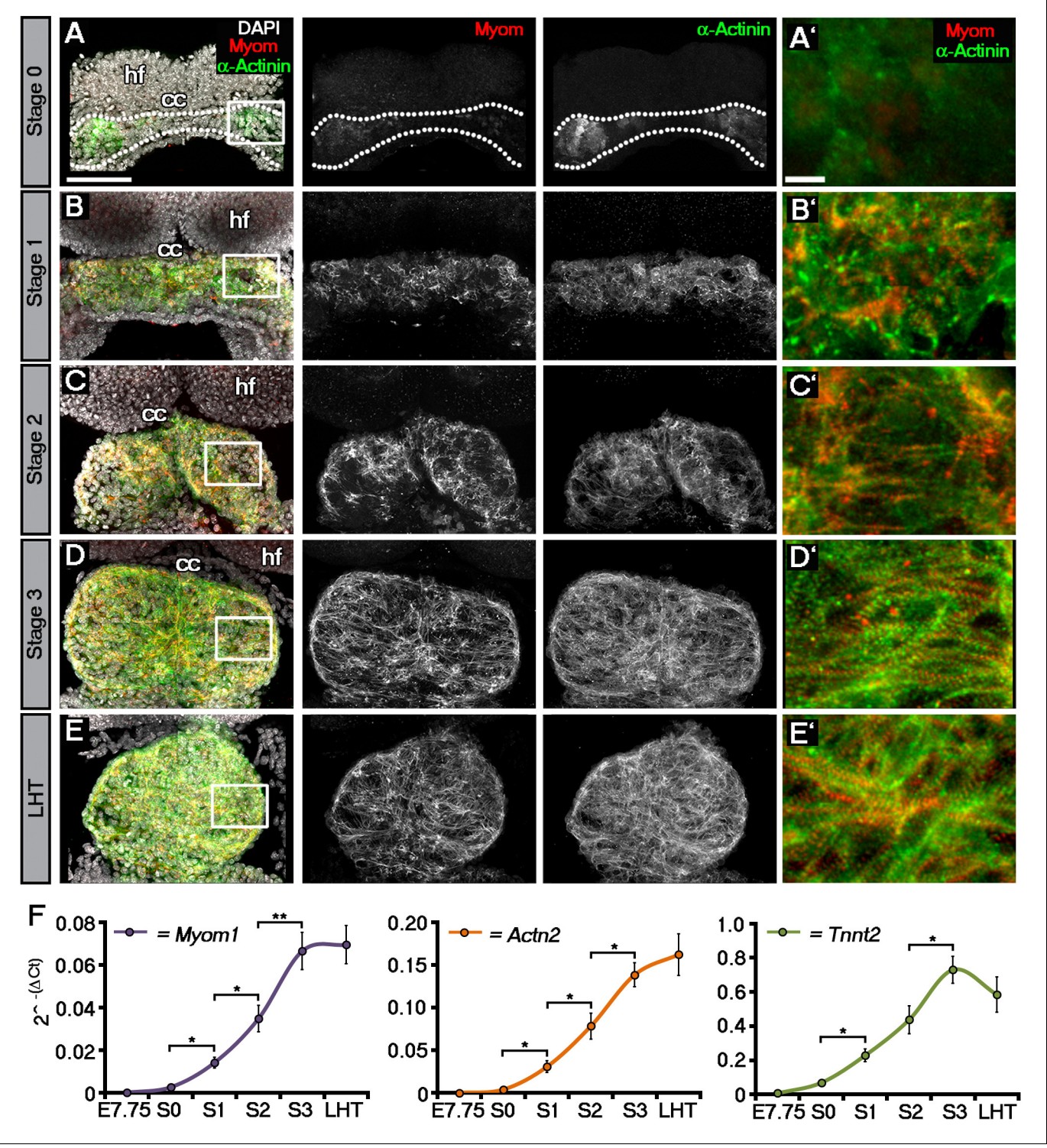

**Figure 1.** Sarcomeric assembly occurs in the forming cardiac crescent during heart development. Maximum intensity projections of alternating myomesin (Myom) and sarcomeric alpha-actinin (α-Actinin) immunostaining from cardiac crescent formation to the linear heart tube stage (LHT; A–E; A, 11 stacks; B, 36 stacks; C, 35 stacks; D, 31 stacks; E, 36 stacks). Analysis by qRT-PCR revealed a significant increase in the expression of *Myom1, Actn2, Tnnt2* (encoding Myomesin, sarcomeric alpha-actinin and cardiac troponin t), in isolated cardiac crescents between stage 0 and stage 1 (F). cc, cardiac crescent (lateral plate mesoderm); hf, head folds (neural ectoderm). Scale bars: A–E, 100 μm, A'–E', 10 μm. Statistics: ANOVA and Tukey test for multiple comparisons (*p<0.05; **p<0.01; ***p<0.001).

*Figure 1 continued on next page*

*Figure 1 continued*

The following figure supplement is available for figure 1:

**Figure supplement 1.** Sarcomeric assembly occurs in the forming cardiac crescent during heart development.

*Tnnt2* expression significantly increased between stages 0 and 1 and continued to increase until formation of the LHT (*Supplementary file 1b*).

## The onset of $Ca^{2+}$ handling in the cardiac crescent

Coincident with sarcomere formation at stage 1 (*Figure 1B*; *Figure 1—figure supplement 1B*), we observed the onset of beating in discrete foci in the lateral regions of cardiac crescents of cultured mouse embryos, by differential interference contrast (DIC) imaging (*Video 1*), significantly earlier than previously described (*Navaratnam et al., 1986*; *Nishii and Shibata, 2006*). These foci generally contracted at the same rate, indicative of either synchronization or a shared intrinsic beat rate for nascent cardiomyocytes (*Figure 2—figure supplement 1*). The earliest cardiac contractions had a frequency of approximately 30 beats per minute (BPM), which increased significantly by stage 3 (after approximately 5 hr) to around 60 BPM (*Figure 2A*). Contractile activity requires cytosolic $Ca^{2+}$ flux, therefore, to investigate the earliest manifestations of $Ca^{2+}$ handling within the cardiac crescent, we loaded embryos with the fluorescent $Ca^{2+}$ indicator Cal-520 followed by live imaging using confocal microscopy. From stage 1 crescents until late stage 3, lateral propagation of $Ca^{2+}$ transients was observed across the embryonic midline (*Figure 2B*), even through regions of non-contractile tissue (*Video 2*). As the cardiac crescent starts to fuse to form the LHT, the transients switch from a lateral to a more caudal-rostral propagation (data not shown). At stage 0 spontaneous asynchronous $Ca^{2+}$ oscillations (SACOs) were observed within the forming crescent, in the absence of contractile activity and sarcomeric banding of cTnT (*Figure 2C*; *Video 3*). SACOs were observed in all stage 0 cardiac crescents imaged (n = 35), propagated within individual cells (*Figure 2D''*) and displayed a range of $Ca^{2+}$ dynamics with variable frequencies and durations (*Figure 2D'*, *Video 4*). Compared to $Ca^{2+}$ transients at later stages (*Figure 2A*) SACOs were significantly slower, with fluorescence reaching peak intensity between 0.79 and 11.9 s and decreasing with a similar slow rate of efflux (*Figure 2—figure supplement 1C*). During a ~20 s recording period we observed only 10.3 ± 0.7 individual SACOs per embryo (n = 35) occurring in different sites. Consecutive SACOs in the same site were rarely observed within the 20 s imaging window and, therefore, we conclude that SACOs in individual cells occur at a frequency < 3 bpm. The appearance of $Ca^{2+}$ transients in the embryonic heart prior to beating (*Figure 2A,B*) is consistent with the idea that $Ca^{2+}$ signalling within early cardiac progenitors may be important to promote sufficient differentiation for subsequent contractile function (*Mesaeli et al., 1999*; *Li et al., 2002*).

## $Ca^{2+}$ signalling and contraction in embryonic stem cell-derived cardiomyocytes

We next used an embryonic stem cell (ESC)-derived model of cardiomyocyte development to complement the embryo studies and provide further insight into the stages of cardiac contraction coincident with cardiomyocyte specification and

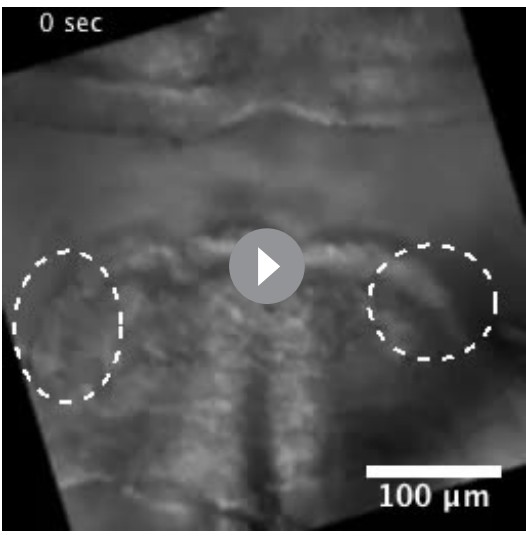

**Video 1.** Representative movie of beating regions in the cardiac crescent at Stage 1. DIC imaging of a stage 1 embryo highlighting beating regions in the lateral regions (dotted circle) of the developing cardiac crescent. Acquisition was performed at 10 frames per second (fps) with a 20x objective and movie played at 8 fps. Scale bar: 100 μm.

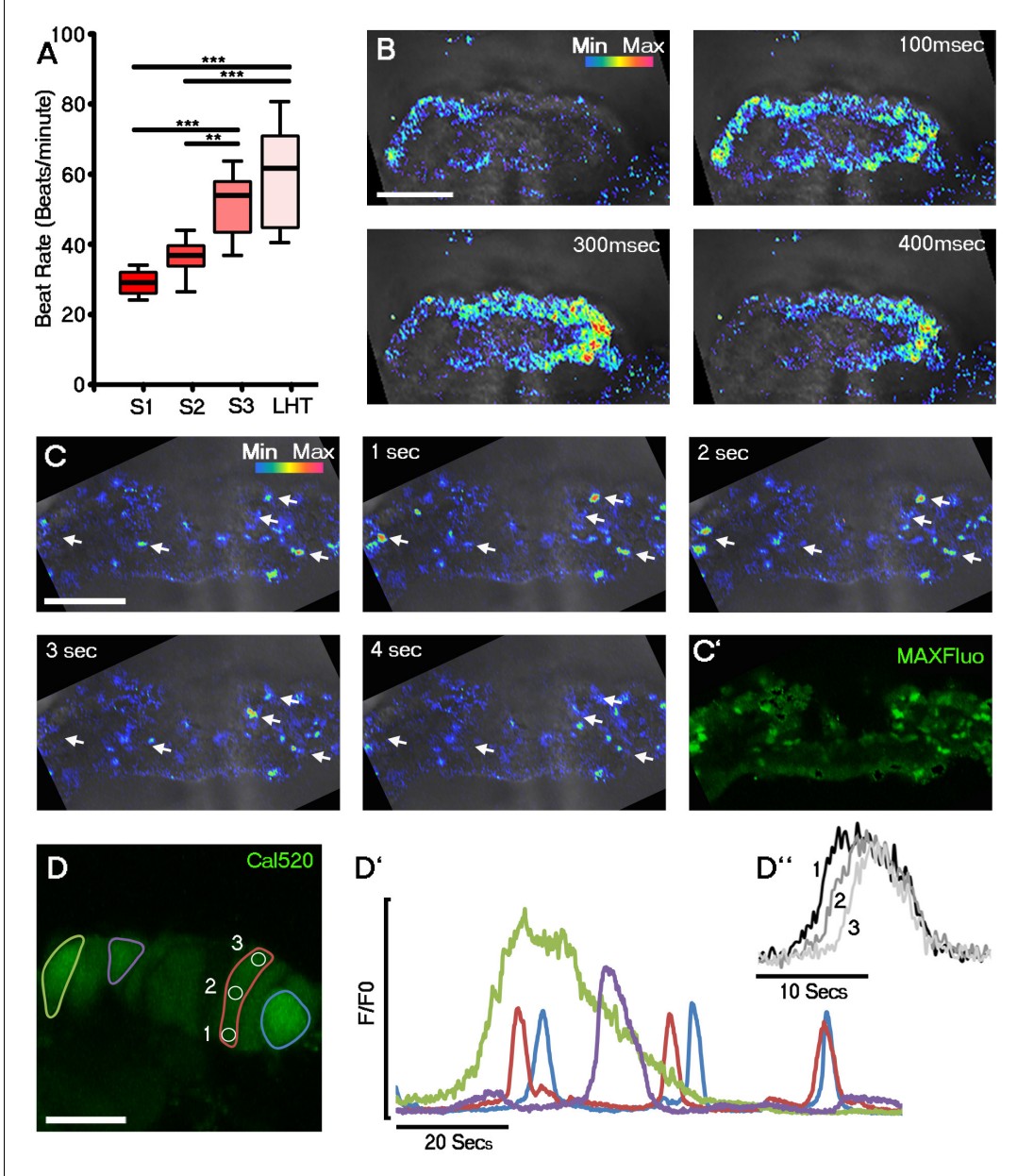

**Figure 2.** Initiation of contraction begins within the forming cardiac crescent and is preceded by spontaneous asynchronous $Ca^{2+}$ oscillations during heart development. Quantitative analysis from the onset of cardiac contraction at stage 1 of crescent formation to formation of the LHT (see *Figure 1* and *Supplementary file 1a* for morphological staging. Stage 1, n = 12; stage 2, n = 8; stage 3, n = 10; LHT, n = 7), revealed a significant increase in heart rate from stages 2 to 3 (A). $Ca^{2+}$ signal following Cal520 loading of stage 1 embryos revealed lateral propagations of transients across the crescent that correlated with the onset of beating at stages significantly earlier than previously described (B). $Ca^{2+}$ signal following Cal-520 loading of stage 0 embryos revealed spontaneous asynchronous $Ca^{2+}$ oscillations in individual cells prior to beating, highlighted by white arrows (C), temporal maximum intensity projection of Cal-520 fluorescence (MaxFluo) over a period of 30 s. Higher resolution imaging of stage 0 single cell $Ca^{2+}$ oscillations represented as a temporal maximum intensity projection over a period of 100 s (D) revealed variation in SACO transient size and frequency (D') and could be observed slowly propagating throughout cells (D''). Scale bars: B, C, 100 µm, D, 20 µm. Statistics: ANOVA and Tukey test for multiple comparisons (*p<0.05; **p<0.01; ***p<0.001).

The following figure supplements are available for figure 2:

**Figure supplement 1.** Discrete foci either side of the embryonic midline beat at the same rate within the stage 1 cardiac crescent.

*Figure 2 continued on next page*

*Figure 2 continued*

**Figure supplement 2.** Principal component analysis of temporal gene expression profiles to cluster embryonic stage with stage of ESC-derived cardiomyocyte differentiation.

**Figure supplement 3.** Cardiomyocyte formation, onset of beating and Ca$^{2+}$ transients are evident by day 7 of ESC-derived cardiomyocyte differentiation.

differentiation. While ESC-derived cardiogenesis is aligned with stages of mesoderm induction, pre-cardiac mesoderm, cardiomyocyte specification and differentiation based on temporal patterns of gene expression (reviewed in *Kattman and Keller, 2007*; *Willems et al., 2009*; *Van Vliet et al., 2012*), this has not been rigorously mapped onto embryonic stages of heart development. We, therefore, performed a principal component analysis (PCA) and hierarchical clustering of 12 cardiac related gene expression profiles of whole embryos across embryonic stages E7.5 to E8.5, compared with ESC-derived embryoid bodies (EBs) from days 0 to 7 inclusive and day 14 of differentiation. Whole embryos were used to reflect the myriad of cell types present in the ESC-cardiomyocyte differentiation assay.

Hierarchical gene clusters were evident with the onset of beating at E7.5 and day 4 and 5 of EB formation, coincident with cardiac progenitor gene expression (*Figure 2—figure supplement 2A*). Expression profiles by days 6 and 7 correlated with E8.0 (stage 0 to stage 2) when beating was well established in both ESC-derived EBs and the embryonic heart, whereas day 14 was equivalent to the more mature E8.5 (*Figure 2—figure supplement 2A*). Key cardiac genes *Myh6, Tnnt2* and *Mef2c* revealed comparable trends of increased expression over time of differentiation in both EBs and embryos (*Figure 2—figure supplement 2*). The cardiac specification program, characterized by expression of *Mesp1*, was evident by day 4 in EBs (*Figure 2—figure supplement 2I*) following *Brachyury* expression, an indicator of mesoderm formation, and loss of pluripotency markers such as *Pou5f1* (encoding Oct-4; *Figure 2—figure supplement 2J–K*). *Mesp1* up-regulation in EBs was consistent with expression at E7.5 in the embryo (*Figure 2—figure supplement 2E*) and preceded beating at day 6 and 7. These later stages revealed an up-regulation of Ca$^{2+}$-handling genes such as *Slc8a1, Cacna1c* and *Ryr2* (*Figure 2—figure supplement 2L–N*).

Sarcomere assembly in ESC-derived cardiomyocytes accompanied the onset of beating at day 7 (*Figure 2—figure supplement 3A,B*) and, interestingly, the rate of beating was comparable at the outset (day 7 of differentiation) with that observed in the developing heart at stage 1 (*Figure 2—figure supplement 3C*), consistent with an intrinsic rate for early cardiomyocytes contractions. Furthermore, Ca$^{2+}$ transients were observed in day 7 EBs as both large propagating waves (*Figure 2—figure supplement 3D,F*; *Video 5*) and within small regions of cells prior to beating, similar to that observed in the stage 0 embryonic heart (*Figure 2—figure supplement 3E,G*; *Video 6*).

## Ion channels are expressed during the earliest stages of heart development

ECC components, and specifically NCX1, have not previously been implicated in the initiation event of cardiac contraction, nor investigated at the earliest stages of heart development coincident with the onset of beating. We, therefore, assessed ECC gene expression on embryonic hearts at the different stages defined herein; focusing specifically on *Slc8a1* (encoding NCX1), *Cacna1c* and *Cacna1d* (encoding the LTCC subunits, Ca$_V$1.2 and Ca$_V$1.3 respectively; *Figure 3A*), *Atp2a2* (encoding SERCA2), *Iptr2* (encoding InsP$_3$ type 2 channels) and *Ryr2* (encoding RyR2) that

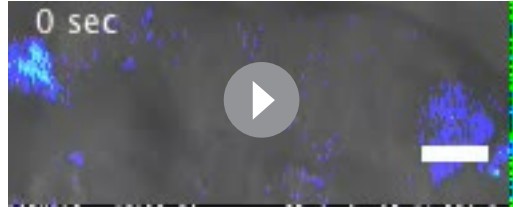

**Video 2.** Representative movie of a Ca$^{2+}$ transient at Stage 1. Confocal time-lapse of a stage 1 embryo loaded with Cal-520. Cal-520 emission (rainbow) was captured simultaneously with DIC imaging (gray). Embryo the same as that shown in *Figure 2D*. Acquisition was performed at 10 fps with a 40x water immersion objective. Background fluorescence was removed by subtracting the signal at a resting phase. Movie played at 8 fps to better show the propagation of the Ca$^{2+}$ transient. Scale bar: 100 µm.

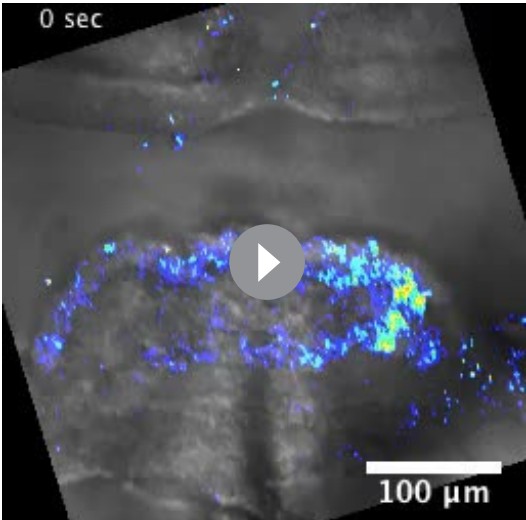
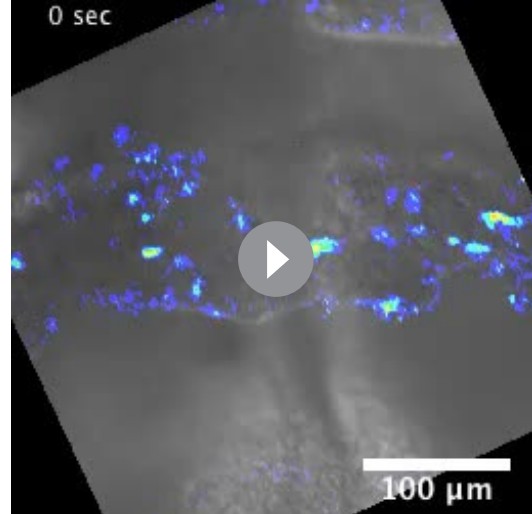

**Video 3.** Representative movie of SACOs at Stage 0. Confocal time-lapse of a stage 0 embryo loaded with Cal-520. Cal-520 emission (rainbow) was captured simultaneously with DIC imaging (gray). Acquisition was performed at 10 fps with a 20x objective. Background fluorescence was removed by subtracting the signal at a resting phase. Scale bar: 100 µm.

**Video 4.** Representative high-resolution movie of SACOs at Stage 0. Confocal time-lapse of a stage 0 embryo loaded with Cal-520. Cal-520 emission (rainbow) was captured simultaneously with DIC imaging (grey). Acquisition was performed at 10 fps with a 20x objective. Movie playback is at 5x original speed. Background fluorescence was removed by subtracting the signal at a resting phase. Scale bar: 10 µm.

collectively are key components of SR $Ca^{2+}$ regulation (*Figure 3—figure supplement 1A–D*; *Supplementary file 1b*). Between E7.75 (as defined by the presence of clear head-folds but not a cardiac crescent) and stage 0, *Slc8a1* significantly increased 21 fold (p-value<0.001), whilst *Cacna1c* revealed a modest but non-significant increase of 2.5 fold (p-value>0.05) and *Cacna1d* a significant 4-fold increase (p-value <0.01). From stage 1 onwards *Slc8a1, Cacna1c* and *Cacna1d* expression continued to significantly increase until stage 3 (*Figure 3A,B*; *Figure 3—figure supplement 1A*, *Slc8a1*, 234-fold; *Cacna1c,* 38-fold; *Cacna1d,* 41-fold). Between stage 3 and the LHT, expression of *Slc8a1* and *Cacna1c* was maintained whilst *Cacna1d* significantly decreased (*Supplementary file 1b*; p-value<0.05). Previous studies have suggested a non-ECC dependent role for the SR during development, implicating both InsP$_3$ and RyR channels (*Méry et al., 2005*; *Sasse et al., 2007*; *Rapila et al., 2008*). While *Ryr2* significantly increased 30 fold between E7.75 and stage 0 (p-value<0.001), expression of *Atp2a2* only increased 1.7 fold (p-value<0.05) and *Iptr2 did not* significantly change (p-value>0.05; *Figure 3—figure supplement 1C,D*).

We performed a similar analysis on EB derived cardiomyocytes. *Slc8a1* expression increased significantly in EBs prior to *Cacna1c* (day 2 versus day 4; *Figure 3B,C*) and to a much greater extent with the onset of beating (110 fold versus 38 fold; *Figure 3B,C*), suggesting NCX1 might play a more immediate role in the onset of beating. This was supported by the lack of any further increase in *Slc8a1* expression from E9.5 through to birth (P0) and adulthood (*Figure 3—figure supplement 2*), whereas *Cacna1c* expression fluctuated across later developmental stages and significantly increased post-natally (*Figure 3—figure supplement 1A,B*) with increased maturation.

Since the sarcolemmal channel genes revealed the most significant increases over the developmental timecourse, we proceeded to investigate spatiotemporal protein expression of NCX1 and Ca$_V$1.2 in embryos (*Figure 3D,E*), and ESC-derived cardiomyocytes (*Figure 3F,G*). Whilst NCX1 was clearly detectable within the cardiac crescent at stage 0 (*Figure 3D*), Ca$_V$1.2 was absent (*Figure 3E*). Differences in the expression of NCX1 and LTCC prior to beating were not maintained at later stages, after established contraction (stage 2 onwards), when both channels were expressed (*Figure 3H,I*). In EBs at day 7 of culture, both NCX1 and Ca$_V$1.2 were expressed, but whilst NCX1

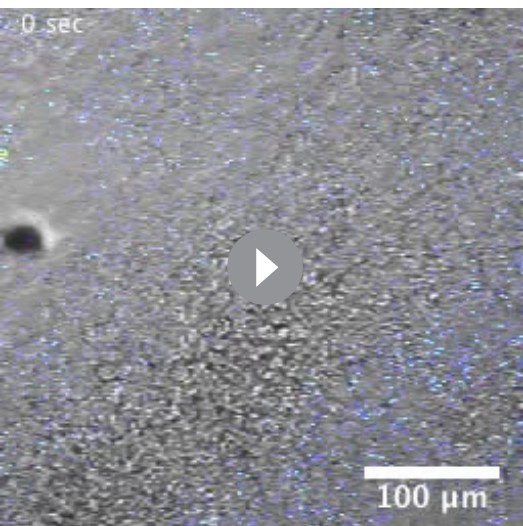

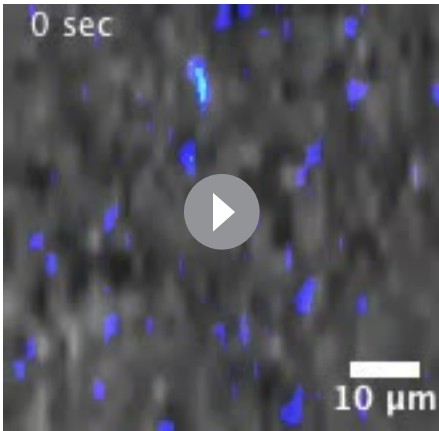

**Video 5.** Representative movie of a propagating Ca$^{2+}$ transient at day eight of ESC cardiomyocyte differentiation. Confocal time-lapse of a day eight EB loaded with Cal-520. Cal-520 emission (rainbow) was captured simultaneously with DIC imaging (grey). Acquisition was performed at 10 fps with a 20x Objective. Background fluorescence was removed by subtracting the signal at a resting phase. Scale bar: 100 μm.

**Video 6.** Representative movie of SACOs at day six of ESC cardiomyocyte differentiation. Confocal time-lapse of a day six EB loaded with Cal-520. Cal-520 emission (rainbow) was captured simultaneously with DIC imaging (grey). Acquisition was performed at 10 fps with a 20x objective. Background fluorescence was removed by subtracting the signal at a resting phase. Scale bar: 10 μm.

was consistently co-expressed with cTnT (*Figure 3F*), there were cTnT+ foci that were negative for Ca$_v$1.2 (*Figure 3G*). Collectively, this expression data suggests that NCX1 precedes Ca$_v$1.2 within the developing cardiac crescent at stage 0, being present prior to and at the onset of both SACOs and beating, whereas Ca$_v$1.2 became upregulated later from stage 1 onwards. Of note, while NCX1 and Ca$_v$1.2 signal was detected in other regions of the embryo, notably in the head folds, in contrast to the heart it was not membrane localised in these regions, indicative of non-functional protein. Furthermore qRT-PCR data for isolated head folds did not reveal any significant increases in *Slc8a1* or *Cacna1c* expression, from E7.75 through to LHT stages (p-value>0.05; *Figure 3—figure supplement 2E,F*; *Supplementary file 1c*) even though the head folds were maturing, as shown by morphological changes and expression of the neural ectoderm marker *Sox1* (E7.75 versus LHT, p-value<0.001; *Supplementary file 1c*)

## Ca$^{2+}$ transients within the forming cardiac crescent are dependent upon NCX1

To functionally assess the roles of the sarcolemmal and SR channels in establishing and maintaining heartbeat, we employed pharmacological blockade on embryos maintained ex-vivo and ESC-derived cardiomyocyte cultures. We inhibited NCX1 using the specific inhibitors CB-DMB (*Secondo et al., 2009*) or KB-R7943 (*Kimura et al., 1999*), the LTCC using nifedipine (*McDonald et al., 1994*) and Ryanodine together with 2-APB to simultaneously block RyR and InsP$_3$, similar to that described previously (*Sasse et al., 2007*; *Rapila et al., 2008*). Treated embryos were imaged for contractile activity by DIC imaging and Ca$^{2+}$ transients were recorded in parallel with confocal fluorescence imaging of Cal-520. Acute treatment (for a maximum of 30 min) with NCX inhibitors (CB-DMB and KB-R7943) or LTCC inhibitor (Nifedipine), affected the embryos in a stage dependent-manner (*Figure 4*). Both NCX1 and the LTCC were required for Ca$^{2+}$ transients during stages 1 and 2 (*Figure 4A*), and their inhibition led to an initial confinement of Ca$^{2+}$ transients to one side of the crescent (within approximately 5 min of treatment, which persisted through to 15 min) followed shortly afterwards by complete loss of Ca$^{2+}$ signal (*Figure 4C,D*). At stage 3 and later, only the LTCC was required for Ca$^{2+}$ transient generation (*Figure 4B*). Whilst NCX1 can function in both forward (Ca$^{2+}$ efflux from the cell) and reverse (Ca$^{2+}$ influx into the cell) modes, our data suggested that NCX1 in the early cardiac

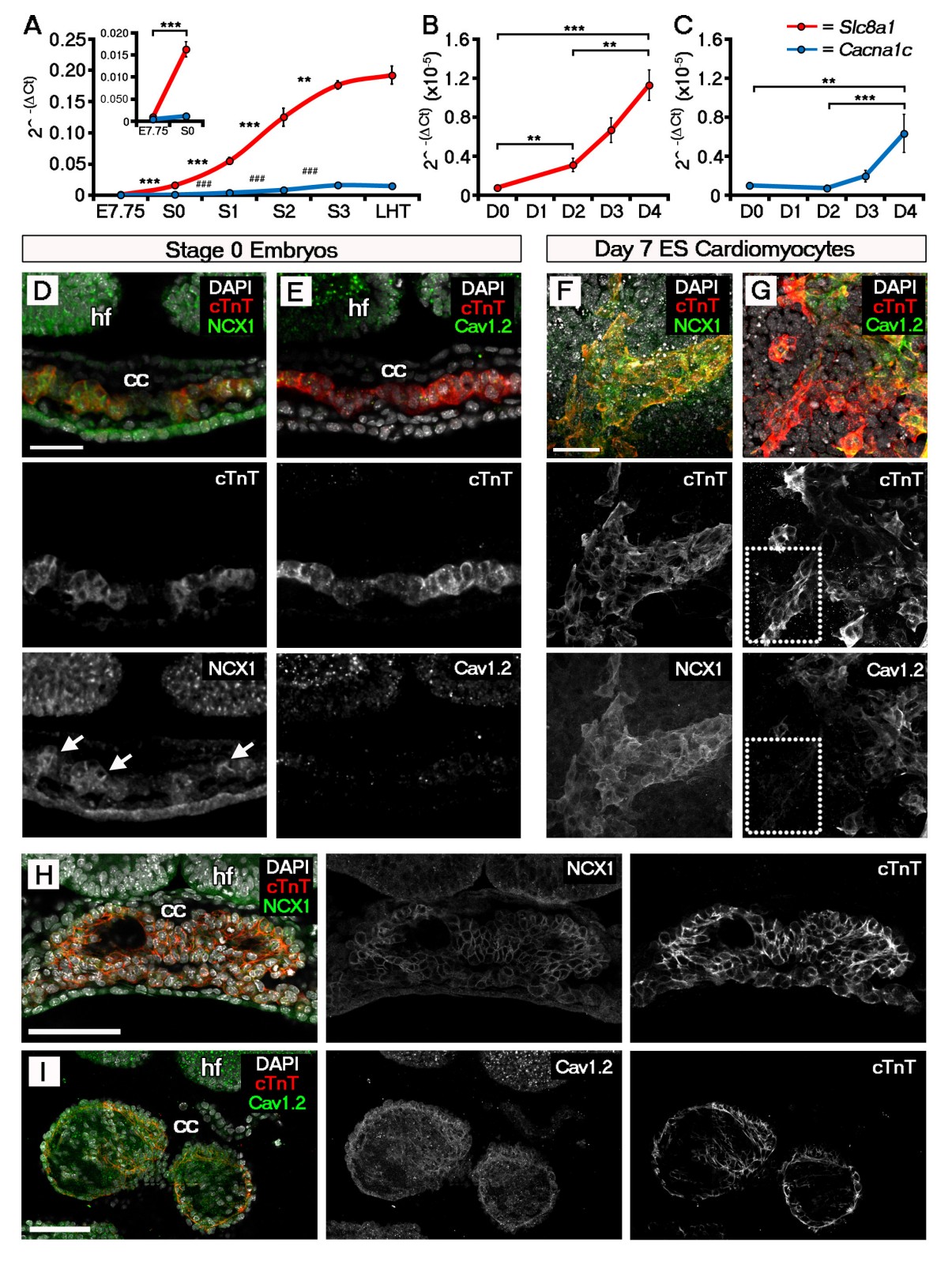

**Figure 3.** The ECC components NCX1 and LTCC are expressed within the early embryonic heart and ESC-derived cardiomyocytes. Analyses by qRT-PCR revealed a significant increase in the expression of *Slc8a1* (encoding NCX1) in the heart from E7.75 to stage 0 (A, n = 5 per stage) and from day 2 of differentiation of ESC-derived cardiomyocytes (B, n = 5 per stage). In contrast expression of *Cacna1c* (encoding the LTCC subunit CaV1.2), increased at a later stages from stage 0 to stage 1 and from day 4 of ESC-derived cardiomyocyte differentiation (C, n = 5 per stage). Confocal imaging section of

*Figure 3 continued on next page*

Figure 3 continued

a stage 0 embryo following immunostaining for cTnT (red) and NCX1 (green), indicated membrane localization of NCX1 within the forming crescent (D; white arrows in lower panel), whereas Ca$_V$1.2 (green) was absent from cTnT+ (red) regions at the same stage (E). A maximum intensity projection of day 7 ESC-derived cardiomyocytes revealed complete overlap of staining for cTnT (red) and NCX1 (green; F, 33 stacks), whereas Ca$_V$1.2 (green) overlapped in part with cTnT (red) but there were also extensive cTnT+/Ca$_V$1.2- regions (dotted box) emphasizing the later requirement for LTCC (G, 22 stacks). Confocal imaging section of a stage 2 embryo following immunostaining for both NCX (H) and Ca$_V$1.2 (I) revealed the expression of both proteins at later stages of heart development. cc, cardiac crescent; hf, head folds. Scale bars: D, F 50 µm, H, I 100 µm. All error bars are mean ± S.E.M; Statistics: one-way ANOVA and Tukey test for multiple comparisons (*p<0.05; **p<0.01; ***p<0.001).

The following figure supplements are available for figure 3:

**Figure supplement 1.** ECC component expression increases during cardiac crescent formation.

**Figure supplement 2.** Expression of *Slc8a1* and *Cacna1c* does not increase in the head folds between E7.75 and the LHT stages of development.

crescent was functioning in reverse mode, as KB-R7943, that is reported to specifically inhibit reverse mode NCX1 function (*Hoyt et al., 1998*; *Iwamoto, 2004*), recapitulated the results with CB-DMB and acted as a control for off-target effects of the latter. These data suggests that reverse mode NCX function may contribute to Ca$^{2+}$ transient generation at the earliest stages of cardiac contraction, potentially via inward Ca$^{2+}$ flux. The inhibitor experiments were repeated on day 7 and day 14 EBs. The relative effects of CB-DMB, KB-R7943 and nifedipine were consistent with those observed in stages 1 and 2 of the embryonic heart (*Figure 4E–G*). Treatment with Ryanodine + 2-APB only affected more mature embryos that had already undergone both cardiac looping and embryonic turning (*Figure 4—figure supplement 1A*) and did not prevent Ca$^{2+}$ transients within cardiac crescents at stage 3 (*Figure 4—figure supplement 1B*). This suggest that a functional SR is only required once cardiac looping has been fully initiated, more than 18 hr later than the first observable SACOs within the cardiac crescent.

Taking into account the earlier upregulation of NCX1 relative to Ca$_V$1.2 at stage 0 (*Figure 3A*), we tested the possibility that NCX1 might be pivotal in initiating the SACOs observed in the early cardiac crescent (*Figure 2C*). Treatment of stage 0 embryos with CB-DMB and KB-R7943 resulted in a 90% decrease in the number of cells with SACOs after drug application when compared to baseline. This was significantly greater than following application of nifedipine to inhibit LTCC, or the DMSO control (CB-DMB, 90% inhibition ± 6.6, mean ± SEM, n = 10 embryos, p-value <0.01; KB-R7943, 91% inhibition ± 7.0, n = 9 embryos, p-value <0.001; nifedipine, 39% inhibition ± 11.4; n = 8 embryos, p-value=0.44; DMSO, 29% inhibition ± 7.6, n = 10; *Figure 4H–J*; *Supplementary file 1d*). These data suggest that NCX1 is required for establishing the earliest pre-contractile SACOs.

## Ca$^{2+}$ channel blockade inhibits ESC differentiation into cardiomyocytes

Chronic treatment of *Eomes*-GFP ESC cultures (ESCs containing a knock-in of a GFP reporter into the endogenous Eomes locus (*Arnold et al., 2009*); marking nascent mesoderm (*Ciruna and Rossant, 1999*), with CB-DMB or Nifedipine, from days 0 to 14 of differentiation, resulted in a reduction in the percentage of beating EBs at day 14, down to 22% and 52% (Percentage beating EBs; DMSO, 76.53% (n = 196); CB-DMB, 22.37%, p-value<0.001 (n = 152); Nifedipine, 52.11%, p-value<0.001 [n = 71]) in the presence of CB-DMB and nifedipine respectively, compared to controls (*Figure 5A*), whilst not effecting cell number (*Figure 5—figure supplement 1A–C*). The reduction in beating EBs was supported by a down-regulation of key cardiac genes *Myh6* (mean+ SEM: 0.57 ± 0.058; p-value<0.05; n = 6; *Figure 5B*) and *Tnnt2* (0.66 ± 0.095; p-value<0.05; n = 6; *Figure 5B*) specifically following treatment with CB-DMB relative to control DMSO treatment. The decrease in cardiac gene expression associated with CB-DMB treatment could be observed from day 4 of *Eomes*-GFP ESC differentiation, at the initiation of cardiomyocyte differentiation (*Figure 5—figure supplement 1D–N*). In an *Nkx2.5*-EGFP ESC line (ESCs containing a vector expressing eGFP under the control of a murine Nkx2.5 promoter and regulatory region; marking cardiac progenitor cells [*Wu et al., 2006*]) this was associated with loss of GFP+ cardiac progenitors by day 14 (*Figure 5—figure supplement 2A–E*) and the significant down-regulation of key cardiac markers following treatment with CB-DMB relative to control DMSO treatment (*Figure 5—figure supplement 2F*). Importantly not all cardiac

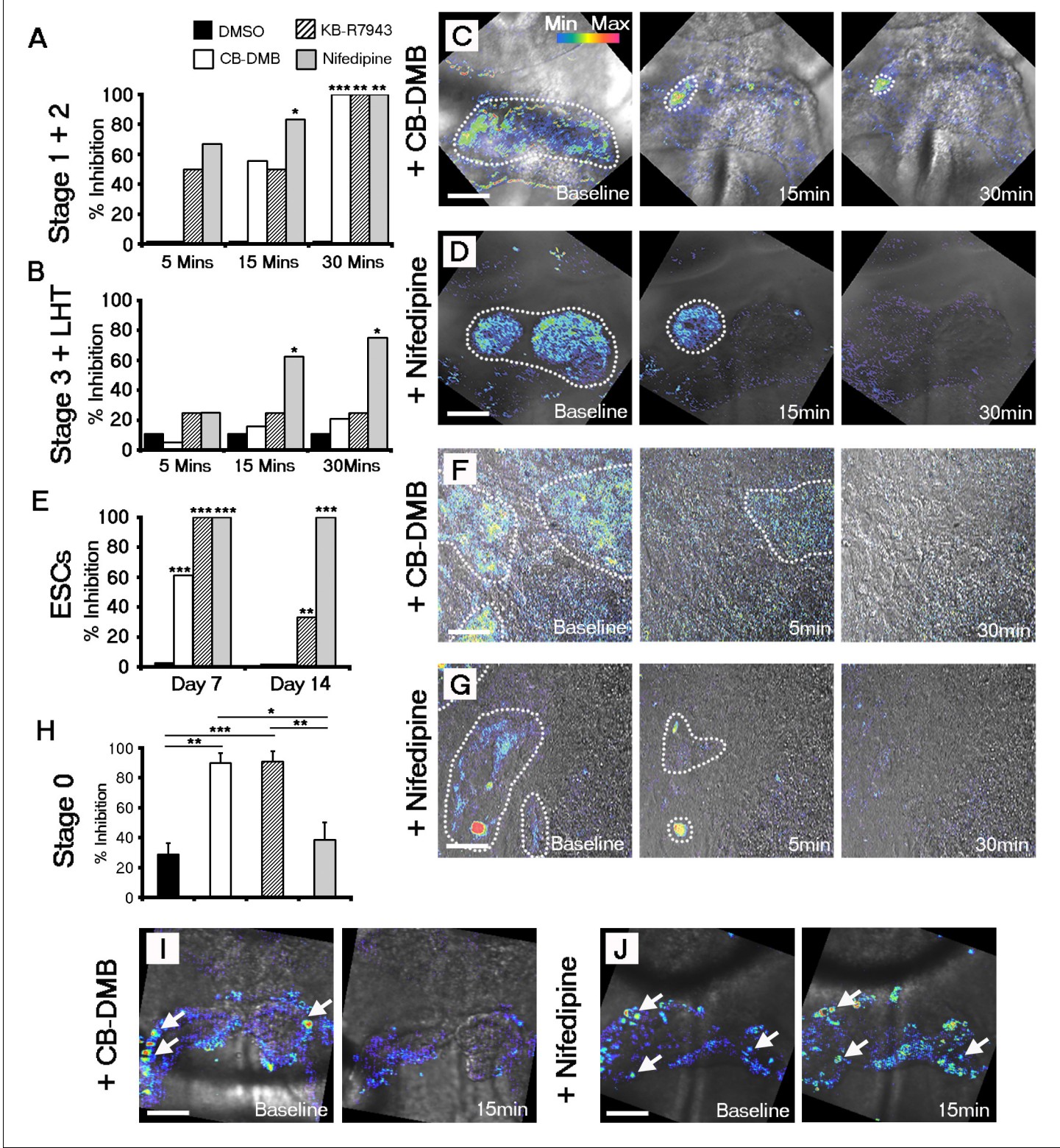

**Figure 4.** Both NCX1 and LTCC are required for Ca$^{2+}$ transients associated with beating cardiomyocytes during cardiac crescent development. Inhibition of Ca$^{2+}$ transients upon treatment of stage 1-LHT embryos with either NCX1 inhibitors CB-DMB, KB-R7943 or the LTCC inhibitor nifedipine, relative to DMSO control after 5, 15 and 30 min of drug application (A, B). Inhibition of NCX1 with either CB-DMB (20 µM) or KB-R7943 (30 µM) affected only stage 1 and 2 embryos (A), whereas inhibition of LTCC with nifedipine (10 µM) effected both stages 1 and 2 and the later stage 3/LHT (B; stage1/2: DMSO, n = 5; CB-DMB, n = 9; KB-R7943, n = 6; nifedipine, n = 6; stage3/LHT: DMSO, n = 9; CB-DMB, n = 19; KB-R7943, n = 8; nifedipine, n = 8). Time

*Figure 4 continued on next page*

Figure 4 continued

series of $Ca^{2+}$ transients on stage1-2 embryos at different time points of either CB-DMB or nifedipine treatment, revealed a confinement to the right side of the embryo prior to complete block (C, D). ESC-derived cardiomyocytes at different days of differentiation were treated with the same channel blockers: inhibition of NCX1 with CB-DMB (10 µM) significantly reduced contractions only in day 7 cardiomyocytes, whereas KB-R7943 (30 µM) affected cardiomyocytes at both day 7 and 14 (E). Inhibition of LTCC with nifedipine (10 µM) significantly reduced contractions in both day 7 and 14 cardiomyocytes and to a much greater extent than KB-R7943 at the later stage (F; day 7: DMSO, n = 38; CB-DMB, n = 36; KB-R7943, n = 7; nifedipine, n = 10; day 14: DMSO, n = 25; CB-DMB, n = 36; KB-R7943, n = 15; nifedipine, n = 15). Time series of $Ca^{2+}$ transients on day 7 ESC-derived cardiomyocytes at different time points of either CB-DMB or nifedipine treatment, revealed a confinement prior to complete block (F, G), equivalent to that observed in the treated embryos (C, D). Treatment of stage 0 embryos prior to the onset of beating with CB-DMB (20 µM) and KB-R7943 (30 µM) resulted in inhibition of slow asynchronous $Ca^{2+}$ transients after 15 min application relative to baseline (H, I; DMSO, n = 10; CB-DMB, n = 8; KB-R7943, n = 9) whereas treatment with nifedipine (10 µM) had no discernible effect on the slow transients (H, J; nifedipine n = 8), supporting the earlier role for NCX1 in initiating $Ca^{2+}$ handling and beating. All scale bars 100 µm. Statistics: Freeman-Halton extension of Fisher exact probability test for embryos for embryos; Chi-square test with Bonferroni correction for ESCs (*p<0.05; **p<0.01; ***p<0.001).

The following figure supplement is available for figure 4:

**Figure supplement 1.** Contribution of the sarcoplasmic reticulum (SR) to $Ca^{2+}$ transients does not occur until looping stages of heart development.

genes were effected by chronic exposure to the channel inhibitors including *Slc8a1, Cacna1c* and *Camk2d* downstream of $Ca^{2+}$ signalling (*Figure 5—figure supplement 3A*), suggesting that the inhibitors did not simply have a global negative effect on gene expression or cardiomyocyte survival.

Given the proposed role for inward $Ca^{2+}$ via NCX1, we cultured ESCs in media containing reduced concentrations of $Ca^{2+}$ (0.1 mM and 1.0 mM) relative to the normal level of 1.8 mM (*Figure 5C*). At the end of the differentiation protocol (14 days) we accessed the percentage of beating EBs after returning to media with normal levels of $Ca^{2+}$ for 2 hr. Culture in 0.1 mM $Ca^{2+}$ resulted in an inhibitory effect on EB beating to the same extent (percentage beating EBs; 1.8 mM, 78.38%, n = 74; 0.1 mM, 28.75%, n = 80; p-value<0.001) as with CB-DMB treatment (*Figure 5C*) and culture in 1.0 mM resulted in an equivalent inhibitory effect to that following treatment with nifedipine (*Figure 5C*; 1.0 mM, p-value<0.01; n = 61). This suggested that influx of external $Ca^{2+}$ is required for cardiomyocyte maturation/beating and supports a potential role for NCX1 working in reverse mode to bring $Ca^{2+}$ into early cardiac progenitors.

We next investigated whether $Ca^{2+}$ handling prior to, and concurrent with, the earliest contractile function impacted on known $Ca^{2+}$-signalling pathways to activate foetal gene expression, as has been reported during adult pathological hypertrophy (*Molkentin et al., 1998*). Calmodulin-dependent kinase II (CaMKII) is a key component of $Ca^{2+}$ and calcineurin signalling which directly impacts on downstream hypertrophic gene expression by induction of Mef2c (*Wu et al., 2006*; *Molkentin et al., 1998*; *Passier et al., 2000*; *Zhang, 2007*). Following CB-DMB-induced NCX1 inhibition at day 7 of EB differentiation, activated phospho-CaMKII (pCaMKII) levels were significantly reduced compared to either DMSO control or nifedipine treatment (mean ± SEM: DMSO, 1.18 ± 0.13; CB-DMB, 0.69 ± 0.15; nifedipine, 1.13 ± 0.06. DMSO vs. CB-DMB, p-value<0.01; CB-DMB vs. nifedipine, p-value<0.05; n = 3; *Figure 5D,D'*). NCX1 inhibition at day 4, a stage prior to cardiac progenitor specification, had no effect on pCaMKII levels (*Figure 5—figure supplement 3B,B'*). NCX1 inhibition reduced $Ca^{2+}$ influx and decreased activation of CaMKII accompanied by significantly reduced expression of *Mef2c* and *Myh7* relative to DMSO control (mean ± SEM: Mef2c, 0.71 ± 0.13; *Myh7*, 0.55 ± 0.18; p-value<0.05; n = 6) which, whilst previously associated with pathological cardiac hypertrophy, contribute here towards (physiological) cardiomyocyte differentiation (*Figure 5B*). In contrast, nifedipine treatment resulted in decreased Myh7, whereas Mef2c remained unaffected at early stages (*Figure 5B*).

## NCX1 is essential for cardiomyocyte differentiation and cardiac crescent formation

To determine the effect of NCX1 blockade on the subsequent development of the heart, we cultured embryos isolated at E7.25 (the onset of head fold formation and pre-cardiac crescent stage) in the presence of DMSO, CB-DMB or Nifedipine for 12 hr (*Figure 5E*). Whilst the embryos remained viable during the culture period, initiation of heart development was impaired in those treated with

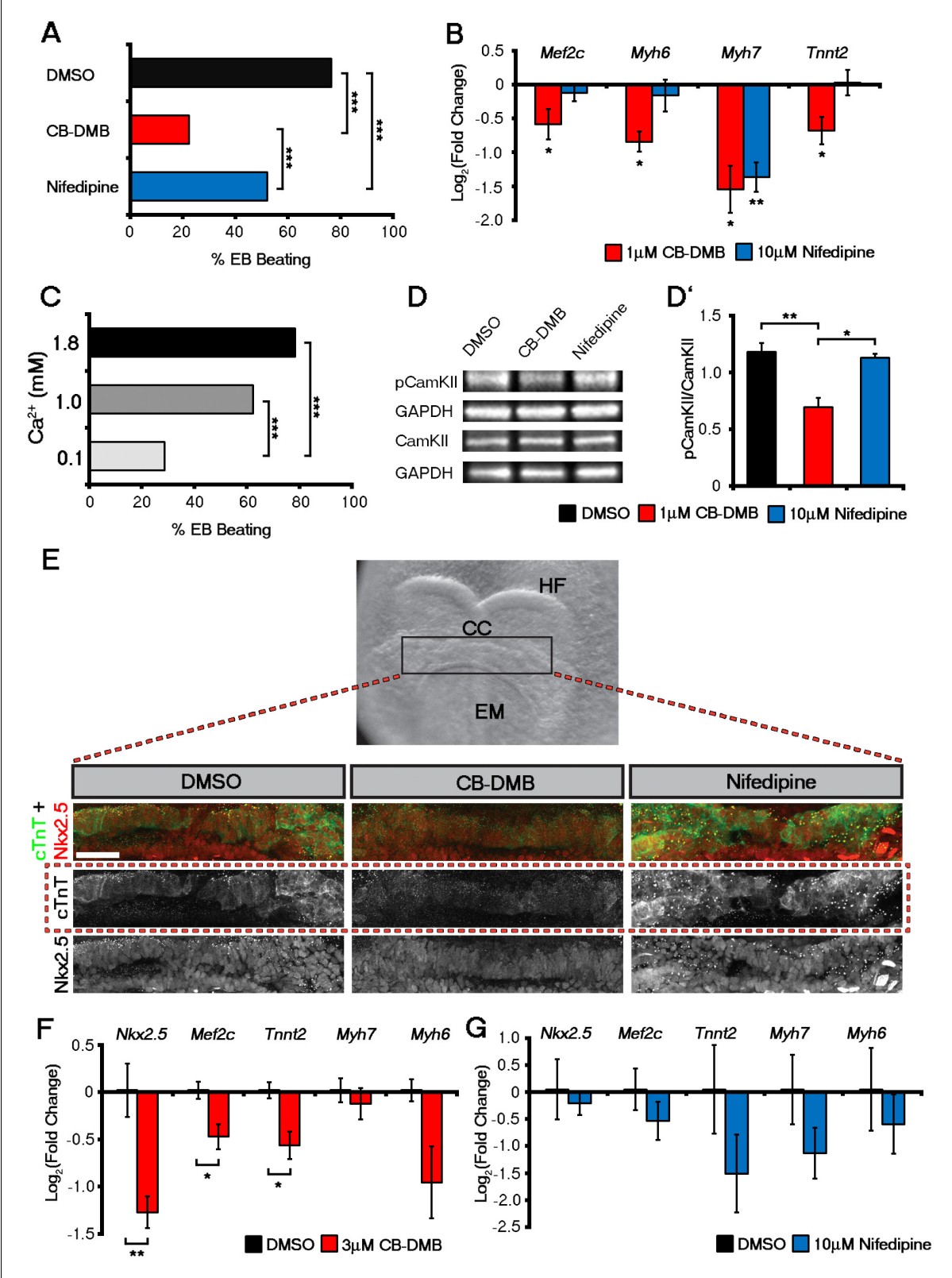

**Figure 5.** Influx of Ca²⁺ and CaMKII signalling are required for early and late cardiac gene expression and crescent formation. Following chronic exposure of embryoid bodies for 14 days to CB-DMB (1 µM) there was a significant decrease in the incidence of beating from 80% to 22% as compared to a reduction to 52% following nifedipine treatment (10 µM) (**A**; DMSO, n = 196; CB-DMB, n = 152; nifedipine, n = 71). Prolonged exposure resulted in a significant decreases in mature cardiomyocyte genes, *Mef2c, Myh6, Myh7* and *Tnnt2* (**B**) gene expression following treatment with CB-DMB (n = 6)

*Figure 5 continued on next page*

*Figure 5 continued*

but not with nifedipine (n = 6). EBs cultured for 14 days in different concentrations of extracellular $Ca^{2+}$ (1.8 mM is the normal culture medium concentration) revealed significantly decreased incidence of beating following culture with reduced $Ca^{2+}$ when assessed in media containing 1.8 mM $Ca^{2+}$ (**C**; 1.8 mM, n = 74; 1.0 mM, n = 61; 0.1 mM n = 80). Cultured ESC derived-cardiomyocytes exposed to NCX1 inhibitors effected downstream $Ca^{2+}$ signalling via alterations in the levels of phosphorylated CaMKII (pCaMKII; **D**). pCamKII to total CamKII ratio was decreased in the presence of 1 μM CB-DMB as compared to 10 μM nifedipine and DMSO (**D'**; n = 3). E7.5 embryos were dissected and cultured for 12 hr in media containing either DMSO, nifedipine (10 μM) or CB-DMB (3 μM) and stained for cTnT and Nkx2.5 (**E**; maximum intensity projections, 30 stacks each). Embryos developed normally in culture, as indicated by head fold formation, coalescence of the cardiac crescent and addition of somites (not shown). Embryos cultured in CB-DMB were delayed in terms of cardiac crescent formation and show a weaker cTnT signal compared to either DMSO alone or nifedipine-treated (**E**; number of affected embryos: DMSO – 1/7; CB-DMB – 7/8; Nifedipine – 1/6). Cultured E7.5 embryos in the presence of either CB-DMB or nifedipine for 12 hr, revealed that CB-DMB significantly down-regulated the expression of both early *Nkx2.5* and *Mef2c* and late *Tnnt2* (**F**, **G**) cardiac genes, coincident with impaired cardiac crescent formation, whereas nifedipine-treatment did not appear to have any effect on cardiac gene expression (**H**). All error bars are mean ± S.E.M. Statistics: **B**, **D**, **G** and **H**: one-way ANOVA and Tukey test for multiple comparisons; **A**, **C**: Chi-square test with a Bonferroni correction for multiple comparisons (*$p<0.05$; **$p<0.01$; ***$p<0.001$). CC, Cardiac crescent; HF, Head folds; EM, Embryonic midline. All scale bars 50 μm.

The following figure supplements are available for figure 5:

**Figure supplement 1.** Inhibition of NCX1 or Cav1.2 did not overtly effect embryoid body formation or cell outgrowth during Eomes-GFP ESC differentiation.

**Figure supplement 2.** NCX1 blockade from the outset of ESC differentiation reduces the incidence of Nkx2.5+ cardiac progenitors.

**Figure supplement 3.** Late administration of NCX1 and LTCC inhibitors did not affect gene expression in ESC-derived cardiomyocytes whereas NCX1 inhibition reduced *Slc8a1* in E7.5 embryos *ex vivo*.

CB-DMB compared to DMSO controls (*Figure 5E*). This was in contrast to treatment with Nifedipine alone, in which the crescent developed normally and progressed to an equivalent stage as DMSO treated embryos (*Figure 5E*). More specifically embryos cultured in the presence of CB-DMB had reduced expression of cTnT compared to embryos cultured in other conditions. However, most cells were still positive for Nkx2.5, suggesting delayed or impaired differentiation likely accounted for the failure of crescent formation under conditions of NCX1 blockade. The relative effects of NCX1 and LTCC inhibition on developmental progression and crescent formation were supported by corresponding gene expression data from cultured embryos (*Figure 5F,G*). While *Tnnt2* was down-regulated in both CB-DMB and Nifedipine treated embryos, *Nkx2.5* was only down-regulated in CB-DMB treated embryos (*Figure 5F,G*). Equally *Slc8a1*, was down-regulated exclusively in the presence of CB-DMB (*Figure 5—figure supplement 3C*), whereas Cacna1c was unaltered in the presence of either inhibitor (*Figure 5—figure supplement 3C,D*). These data collectively support a role for NCX in cardiomyocyte differentiation and crescent formation.

## Discussion

Previous studies have attempted to investigate how cardiac function develops within the early embryo. Whilst these studies are informative they rely on the dissociation and culture of embryonic cardiomyocytes to facilitate physiological measurements resulting in the loss of critical spatial and temporal information regarding $Ca^{2+}$-handling and downstream changes in gene expression and morphology. Using a staging method based on morphological landmarks (stages 0–3), we characterized in detail the in vivo progression of physiological activity during early heart development. The stage series defined correlates with gradual expression of several cardiac related genes and sarcomere assembly. We observed spontaneous asynchronous $Ca^{2+}$ oscillations (SACOs) at stage 0 in the developing cardiac mesoderm, before any detectable cardiac contractions. These transients appeared sporadically in individual cells within the forming cardiac crescent and did not appear to be synchronized. At stage 1, periodic $Ca^{2+}$ transients began to be propagated laterally through the cardiac crescent, and traversed the midline of the crescent where there were no visible contractions. This was a rapid and dynamic process occurring through stages 0–3 that involved sarcolemmal ion channel and exchanger function. NCX1 was exclusively required for SACOs at stage 0, whereas both

NCX1 and LTCC appeared to play equally important roles at stage 1 and 2. NCX1 was no longer required from stage 3 onwards when the LTCC channels maintained $Ca^{2+}$ transients. We observed no contribution of the SR in regulating $Ca^{2+}$ transients within the cardiac crescent, as assessed by simultaneous blockade of RyR2 and InsP$_3$. However, consistent with previous studies (*Sasse et al., 2007*; *Rapila et al., 2008*), our data did show that the SR becomes functional at later stages during cardiac looping (~E8.5). This may represent a period in which SR $Ca^{2+}$ filling is required before a threshold for $Ca^{2+}$ release from the SR can occur, as has been proposed in the adult setting (*Stokke et al., 2011*). Overall our data reveals that during the time frame from the earliest cardiac morphogenesis (crescent maturation) to looping of the heart (approximately 18 hr) there is a rapid transition in the mechanism by which the intracellular concentration of $Ca^{2+}$ is elevated, based on distinct channel and exchanger function.

The observation of spontaneous asynchronous calcium oscillations (SACOs) within the forming cardiac crescent was a surprising finding that, to the best of our knowledge, has not previously been reported in any type of excitable cell. The specific role of SACOs is currently unknown and we hypothesise that they are required in cells that need to optimally activate $Ca^{2+}$-dependent signalling (via the CAMKII pathway) in order to up-regulate genes necessary for further differentiation and morphogenesis. This would explain why SACOs are present much earlier than complete sarcomere assembly. An alternative hypothesis is that SACOs are a by-product of cells that are already committed to specific cardiac lineages and, therefore, arise with the expression of specific channels required for future function. This could explain the variation in duration and frequencies of SACOs observed within the same embryo. At this point, quite how SACOs become synchronised transients is unknown. We speculate that release of a $Ca^{2+}$-dependent signal from a 'pacemaker' cell may entrain neighbours to have synchronised transient periodicity. To this end we observed highly variable $Ca^{2+}$ periodicity but also regions containing cells of similar periodicity (*Video 2*). Since blockade of NCX1 prior to the formation of the cardiac crescent, and chronic treatment in ESCs, leads to impaired cardiac differentiation it is also possible that SACOs may be present in mesoderm cells earlier than reported here, which we were not able to image due to the limitations of the current experimental set-up.

In the embryonic heart, $Ca^{2+}$ handling is assigned to sequential roles for the NCX and LTCC channels, whereby NCX has been assumed to compensate, at least in part, for the rudimentary (non-functional) SR in the developing mouse heart (*Conway et al., 2002*). In zebrafish embryos it has recently been demonstrated that $Ca^{2+}$ handling and not contraction per se is essential for regulating cardiomyocyte development (*Andersen et al., 2015*). Previous studies characterised functional expression of NCX from E8.5-E9.5 (post-LHT formation) in mouse (*Linask et al., 2001*; *Reppel et al., 2007*) coincident with cardiac looping and significantly later than the onset of the first heat beat described herein. NCX1 knock-out mice have been generated by multiple groups, with conflicting results in regards to the phenotype, extent of mutant heart development and the stage at which embryonic lethality occurs (*Wakimoto et al., 2000*; *Koushik et al., 2001*; *Reuter et al., 2002*; *Cho et al., 2003*). This suggests that loss of NCX1 may be compensated for at the level of early cardiomyocyte specification, differentiation and contraction. In mammals there are three different $Na^+$-$Ca^{2+}$ exchangers (NCX1, NCX2 and NCX3), and it has been previously demonstrated that these three exchangers share similar physiological properties (*Linck et al., 1998*; *Lytton, 2007*). Furthermore, NCX1 has several splicing variants with exon 1 being mutually exclusive to exon 2 (*Lytton, 2007*; *Quednau et al., 1997*). It is, therefore, possible that either a different NCX or an alternative splice variant will compensate for loss of the NCX1 variant targeted in the previous studies. Indeed, in all of the previous NCX1 generic loss-of-function studies, mutant mice were created by targeting exon 2, supporting the possibility that an alternately spliced variant may be able to compensate. Furthermore, precedent for genetic ablation being compensated for over the course of development exists, whereby patterning defects were muted over time (*Bloomekatz et al., 2012*). In contrast, our use of pharmacological channel blockers CB-DMB and KB-R7943, resulted in an acute stage-specific-loss of NCX1 function which attributed an essential role for NCX1 in generating SACOs, and potentially acting as a trigger to establish the onset of beating in the cardiac crescent and subsequent cardiomyocyte differentiation and morphogenesis. These findings are consistent with the situation in tremblor (*tre*) zebrafish which have a mutation in NCX1 leading to absence of normal calcium transients and cardiac fibrillation (*Langenbacher et al., 2005*) as well as earlier findings in rodents demonstrating a

requirement for elevated cytoplasmic calcium to drive cardiac myofibrillogenesis in developing cardiomyocytes (*Webb and Miller, 2003*).

Parallel analyses on ESC-derived cardiomyocytes revealed that lowering the concentration of $Ca^{2+}$ in the media had a similar effect on reducing the number of beating EBs as treatment with CB-DMB. This both reinforced the relative importance of NCX1 and also provided the first indication that the exchanger may be working in reverse mode, facilitating inward $Ca^{2+}$ as necessary for cardiomyocyte differentiation. Furthermore, our use of the inhibitor KB-R7943, previously reported to specifically inhibit reverse mode NCX1 activity (*Brustovetsky et al., 2011*), replicated observations made with CB-DMB, such that at stage 1 and 2 both LTCC and NCX1 are required for sufficient $Ca^{2+}$ influx. Whilst inhibition of NCX1 clearly blocked SACOs in stage 0 cardiac crescents, the mechanism by which NCX1 alone could lead to periodic oscillations in $Ca^{2+}$ is still unclear, and may require oscillations in other ions, such as $Na^+$, and/or contribution of other sarcolemmal proteins, such as the Plasma membrane $Ca^{2+}$ ATPase (PMCA), to regulate $Ca^{2+}$ efflux while NCX1 is working in reverse mode. Due to the slow nature of SACOs, oscillations in energy production along with adenosine triphosphate levels could also be involved in SACO generation, especially with reduced SR function. Whilst SR inhibition did not prevent stage 3 $Ca^{2+}$ transients, we have not tested the involvement of SR function at stage 0 and, therefore, cannot fully exclude that periodic $Ca^{2+}$ releases from the SR are involved in the generation of SACOs. The latter has been reported in cultured E8.5–9 embryonic cardiomyocytes as a pace making mechanism (*Sasse et al., 2007*; *Rapila et al., 2008*) at later stages than were the focus in this study. To further understand the mechanism of NCX1 in SACO generation will require *in vivo* electrophysiological analysis to more accurately determine whether the inhibitory effect of NCX1 blockade with CB-DMB and KB-R7943 is due to effects on reverse mode NCX1 function (inhibition of translemmal $Ca^{2+}$ influx) or forward mode NCX1 function (inhibition of $Ca^{2+}$ efflux leading to $Ca^{2+}$ overload), imbalance of $Ca^{2+}$ homeostasis or off-target effects. Although the existence of the NCX1 acting in reverse mode is contentious, and the precise function of KB-R7943 is still debated in the field, it is difficult to otherwise explain how inhibition of NCX1 can block SACOs. That said, regardless of mode of action, the observation that at stage 0 SACOs were abolished with both CB-DMB and KB-R7943 treatment, but persisted when treated with nifedipine, suggests that NCX1, and not the LTCC, plays a major role in $Ca^{2+}$ transient generation within the early cardiac crescent.

As cardiomyocytes matured, transition to LTCC became the predominant mechanism for inward $Ca^{2+}$ entry, as demonstrated by nifedipine-induced inhibition of beating at later stages in both the embryo and ESC-derived cardiomyocytes. NCX1 expression was further maintained at high levels during more advanced stages (stage 0-LHT), suggesting a later role in ensuring $Ca^{2+}$ removal via its forward mode of action. Of note, the early versus late roles for mammalian NCX1 and LTCC, respectively, are further supported by studies on *tre* zebrafish, whereby sarcomeric assembly defects in developing cardiomyocytes following loss of NCX1 function are not recapitulated by mutations in LTCC (*Ebert et al., 2005*). Increased expression of NCX1 has also been linked with pathological hypertrophy and heart failure (reviewed in *Sipido et al., 2002*), whereby elevated NCX1 is thought to compensate for defective ECC and depressed function of SERCA but also produce arrhythmogenic-delayed after-depolarisations (*Gómez et al., 1997*; *Schultz et al., 2004*; *Venetucci et al., 2007*). Transgenic mice over-expressing NCX1 within the myocardium exhibited a proportional decrease in contractile function and increased incidence of heart failure, suggesting a decompensatory mechanism with regards to $Ca^{2+}$ handling (*Roos et al., 2007*).

Delta isoforms of CaMKII predominate in the heart and are involved in multiple signalling cascades to regulate gene expression, as well as cardiomyocyte physiology including $Ca^{2+}$ and $Na^+$ homeostasis (*Wagner et al., 2006*; *Aiba et al., 2010*). In this study, CaMKII activation was impaired following NCX1 inhibition in embryonic stem cells, and, moreover, canonical hypertrophy genes, including *Mef2c* and *Myh7* were down-regulated in response to impaired NCX function during development. Whilst changes in gene expression did differ between embryos and ESCs in response to treatment with different drugs, this likely reflects inherent differences between the in vitro versus *in vivo* models, as well as the timing and length of drug exposure. ESCs were exposed at a relatively earlier stage of cardiac lineage specification and for longer periods, which was not feasible in the live embryo cultures, resulting in a stronger down-regulation of the genes investigated. Overall though these findings suggest analogous roles for NCX1 in early heart development and adult

hypertrophy, with regards $Ca^{2+}$-handling and foetal cardiac gene induction to promote physiological or pathological myocyte growth, respectively.

There is recent precedent for a role for $Ca^{2+}$ in the establishment of other embryonic lineages; most notably $Ca^{2+}$ signals are involved in the earliest steps of neurogenesis, including neural induction and the differentiation of neural progenitors into neurons (*Leclerc et al., 2012*). Here we show that pharmacological inhibition of NCX1 and dysregulation of $Ca^{2+}$ handling from the outset had an adverse effect on early cardiomyocyte differentiation and led to impaired cardiogenesis in the embryo (*Figure 5*). Thus, an early induction of $Ca^{2+}$-handling preceding beating within cardiac muscle is pivotal for subsequent terminal differentiation and normal heart development.

## Materials and methods

### Mouse strains, husbandry and embryo collection

All animal experiments were carried out according to UK Home Office project license PPL 30/3155 and 30/2887 compliant with the UK animals (Scientific Procedures) Act 1986 and approved by the local Biological Services Ethical Review Process. To obtain wild-type embryos C57BL/6 males (in house) were crossed with CD1 females (Charles River, England). All mice were maintained in a 12-hr light-dark cycle. Noon of the day finding a vaginal plug was designated E0.5. In order to dissect the embryos the pregnant females were culled by cervical dislocation in accordance with the schedule one of the Animal Scientific Procedures Act. Embryos of the appropriate stage were dissected in M2 medium (Sigma-Aldrich, England).

### Embryo staging

Progressive crescent stages were defined based on morphological criteria: the length (medio-lateral axis) and the maximum height (rostral-caudal axis) of the cardiac crescent was measured for each embryo. Embryos were considered to be at the LHT stage once both sides of the cardiac crescent were completely folded and fused. The ratio between width and maximum height measurements (μm) was used to categorize the embryos from stage 0 through to stage 3 (*Supplementary file 1a*).

### Live imaging – DIC, $Ca^{2+}$, embryos and cells

Live imaging of embryos, including $Ca^{2+}$ imaging was performed as previously described, with some adaptations (*Chen et al., 2014*). Briefly, freshly dissected embryos were imaged in a mix of 50% phenol red-free CMRL (PAN-Biotech, Germany) supplemented with 10 mM L/glutamine (Sigma-Aldrich) and 50% Knockout Serum Replacement (Life Technologies, England). Initial characterisation of cardiac contractions was performed using Differential Interference Contrast (DIC) imaging on a Spinning Disk Confocal microscope at 37°C and an atmosphere of 5% $CO_2$ + Air. Images were acquired at 10 frames per second (fps) for up to 20 s. For all experiments involving $Ca^{2+}$ imaging, embryos were loaded with 8 μM of Cal-520 by incubating the embryos in 50% CMRL + 50% Knockout Serum Replacement with the dye for 15 min at 37°C and an atmosphere of 5% $CO_2$ + Air. The embryos were then transferred to fresh media in a MatTek dish (MatTek Corporation, Framingham, MA) without the dye and imaged. To image $Ca^{2+}$ transients in embryoid bodies (EBs), ESCs were cultured in hanging drops in MatTek dishes for four days under differentiation conditions. At day 7, the cells were loaded with 8 μM Cal-520 for 30 min in media without serum at room temperature and then transferred into media with serum and cultured in the presence of the dye for 10 min at 37°C and an atmosphere of 5% $CO_2$ + Air. All $Ca^{2+}$ imaging was performed with a Zeiss 710 LSM fitted with an environmental chamber to maintain the embryos at 37°C at 5% $CO_2$. Embryos were imaged with a 20× air objective (0.6 NA) with a single optical section every 97 ms (~10 frames per second). Images were captured at 256 × 256 pixel dimensions, with a 2× line step and no averaging to increase the scan speed. For acute inhibition of $Ca^{2+}$ transients in embryos (stage 1 to LHT) and EBs, $Ca^{2+}$ transients were first imaged at baseline, drug containing media was then added and imaging was performed at 5, 15 and 30 min post drug treatment. Inhibition was defined as the complete cessation as well as confinement of $Ca^{2+}$ transients to small regions of the cardiac crescent. For inhibitor experiments involving stage 0 embryos, imaging was carried out at baseline, 5 and 15 min post treatment. For all the experiments involving calcium imaging of embryo the overlying yolk sac endoderm had to be dissected out in order to allow proper drug and dye penetration, as well as better

visualisation of the cardiac mesoderm. For experiments involving embryos Nifedipine (Sigma-aldrich) was used at a final concentration of 10 μM, CB-DMB (Sigma-aldrich) at 20 μM, KB-R7943 (Sigma-aldrich) at 30 μM, Ryanodine (Tocris Bioscience, England) at 100 μM and 2-APB at 200 μM, all diluted in DMSO (Sigma-aldrich). We found that most drugs had to be used at a higher concentration than previously used in studies involving isolated cells, presumably due to penetration difficulties inherent with using whole embryos. Control experiments for Nifedipine, CB-DMB and KB-R7943 were performed with 0.002% DMSO, and for experiments involving dual inhibition with Ryanodine + 2-APB with 0.6% DMSO. All experiments involving embryos were repeated with at least three different litters (6–10 embryos). Both embryo and ESC experiments were performed on at least three independent days.

## Embryo culture

Embryos were cultured in the presence of drug (CB-DMB or nifedipine) for 12 hr, from E7.5 to E8.0 using a rolling culture system. Embryos were cultured in a mix of 50% CMRL and 50% Knockout Serum Replacement at 37°C and with an atmosphere of 5% $CO_2$+ Air. Only embryos at E7.5, prior to cardiac crescent and head fold formation were cultured. For these experiments embryos were cultured in the presence of either in 10 μM Nifedipine, 3 μM CB-DMB or 0.0003% DMSO. Experiments for immunostaining were performed on three independent days, while experiments for qRT-PCR were performed on four different days for Nifedipine treatments and five days for CB-DMB treatments.

## ES cell culture and cardiomyocyte differentiation

Cardiomyocyte differentiation from ESCs was carried out as previously described using an *Nkx2.5*-GFP (*Moretti et al., 2006*) and *Eomes*-GFP (*Arnold et al., 2009*) cell line. Briefly, ESCs were maintained in an undifferentiated state by culturing with KO-DMEM (Gibco, England) supplemented with glutamax (2 mM; Gibco), embryomax FBS (15%; Millipore), nonessential amino acids (0.1 mM; Invitrogen, England), penicillin (60U/mL; Gibco), streptomycin (60 μg/mL; Gibco), β-mercaptoethanol (0.1 mM; Sigma-aldrich) and Leukaemia inhibitory factor (1000 U/ml; Millipore, England). Cardiomyocyte differentiation was induced using the hanging drop culture method (*Kattman and Keller, 2007*). Approximately 500 ESCs were plated in 20 μl drops of differentiation media (4.5 g glucose/DMEM, embryomax FBS (20%), nonessential amino acids (0.1 mM), penicillin (60 U/ml), streptomycin (60 μg/ml), β-mercaptoethanol [0.1 mM]) on the lids of petri dishes and cultured as hanging drops throughout the first four days of differentiation, allowing embryoid bodies (EBs) to form. At day 4, the EBs were transferred onto 0.1% gelatin coated plates for a further 10 days of culture before being collected at day 14 of differentiation. 1 μM CB-DMB and 10 μM Nifedipine were added to the differentiation media at day 0. For control experiments, DMSO was added at the same concentration as drug containing media. Once the EBs had been plated at day 4, drug-containing media was changed every two days up until day 14. To assess the percentage of EBs which were beating, drug containing media was removed 2 hr prior to assessment and replaced with fresh drug-free culture media. Culturing of ESCs in different $Ca^{2+}$ concentrations was achieved using $Ca^{2+}$ free DMEM (Gibco) with $Ca^{2+}$ being added back to the desired concentration (0.1 mM, 1.0 mM, 1.8 mM). For immunostaining, ESCs were cultured on 0.1% gelatin coated glass coverslips before being fixed with 3.7% PFA for 30 min on ice. For RNA isolation tissue samples were dissociated for 2 min using 0.25% trypsin-EDTA at 37°C prior to snap freezing.

## Immunostaining

Dissected embryos were fixed for 1 hr at room temperature with 4% PFA in PBS. The embryos were then washed 3x in PBT-0.1% (PBS with 0.1% Triton X-100) for 15 min, permeabilised in PBT-0.25% for 40 min and washed again 3x in PBT-0.1%. The embryos were transferred to blocking solution (5% donkey serum, 1%BSA in PBT-0.1%) overnight (o/n) at 4°C. Primary antibodies (*Supplementary file 1e*) were then added to the solution and incubated o/n at 4°C. The embryos were washed 3x in PBT-0.1% and incubated o/n 4°C in PBT-0.1% with the secondary antibodies (*Supplementary file 1e*), then subsequently washed 3x PBT-0.1% for 15 min and mounted in Vectashield mounting media with DAPI for at least 24 hr at 4°C. After fixation ESC samples were rinsed with PBS before being permeabilised with PBT-0.1% for 10 min, followed by blocking with 10% goat

serum, 1% BSA in PBT-0.1% for 1 hr. Incubation with primary antibodies, diluted in blocking buffer, was carried out o/n at 4°C. After incubation with primary antibodies samples were washed for 3x for 10 min with PBT-0.1% and then incubated with secondary antibodies (*Supplementary file 1e*) diluted in blocking buffer for 1 hr at room temperature. After incubation with secondary antibodies samples were washed 5x for 5 min in PBS. Samples were mounted using Vectashield mounting media with DAPI for at least 24 hr prior to imaging with either a 40x oil (1.36 NA) or water (1.2 NA) objective. Images were captured at a 512 × 512 pixel dimension and tiled 2x2 with a Z-step of 1.5 µm. For embryos each staining was repeated for at least three litters (6–10 embryos per litter). Both experiments involving embryos and ESCs were performed on at least three independent days and with two different secondary antibodies.

## RNA-extraction and qRT-PCR

RNA extraction of whole embryos, embryonic hearts, head folds and ESC samples was performed using an RNeasy Micro Kit (Qiagen, England) according to manufacturer's instructions: briefly, homogenisation was carried out with a 21G needle and the extract run through an on-column DNase I treatment. For P0 and adult heart samples, RNA extraction was performed using Trizol (Invitrogen, England), according to the manufacturer's instructions, additional DNase I (Promega, England) treatment on Trizol-extracted RNAs was carried out to eliminate genomic DNA contamination. In order to collect enough RNA for isolated cardiac crescent and head fold development, each biological replicate was composed of 10 individual cardiac crescents or head folds. For both types of extraction, RNA pellets were dissolved in RNase-free water and the RNA quality and quantity determined by Nanodrop readings at 260, 280 and 230 nm wavelengths. cDNA was generated from 1 µg of RNA using random primers and SuperScript III polymerase (Invitrogen). The expression of mRNAs for genes of interest (*Supplementary file 1f*), together with endogenous controls (treated embryos and differentiated ES cells, HPRT, GAPDH, 18 s; *in vivo* timecourse, GAPDH & HPRT; ES cell differentiation, GAPDH & 18 s (*Murphy and Polak, 2002*) were measured in triplicate for each sample by quantitative real-time PCR using SYBR Green (Applied Biosystems, England). Each reaction contained: 8 ng cDNA, 0.5 µl of each primer, 6.5 µl water and 12.5 µl 2 x SYBR Green, made up with H$_2$O to a final volume of 22 µl. Primers (Sigma-aldrich) were either designed using Primer-BLAST (National Center for Biotechnology Information, National Institutes of Health) or obtained from PrimerBank (http://pga.mgh.harvard.edu/primerbank; Primer 3) or previous publications (primer sequences in *Supplementary file 1f*). Primers were designed to span exon-intron boundaries, have annealing temperatures around 60°C and generate amplicons between 50–200 bp. The reaction mixture and samples were loaded into either a MicroAmp Optical 96-Well Reaction Plates or MicroAmp Fast Optical 96-Well Reaction Plates and sealed with Optical Adhesive Films (Life Technologies). Quantification was performed on a ViiA 7 Detection System (Applied Biosystems) using a PCR programme of 95°C 15 min followed by 40 cycles of (95°C 15 s melting phase and 60°C 1 min annealing and extension). Amplification of a single amplicon was confirmed by obtaining dissociation curve (melt curve) profiles as well as using gel electrophoresis to separate the reaction product. Cycle threshold (Ct) values were generated using either Viia7 software (Applied Biosystems). Relative gene expression levels were obtained using the ∆∆Ct method, in which expression of each gene of interest was normalised to endogenous controls (*Schmittgen and Livak, 2008*) (*Murphy and Polak, 2002*), and presented as fold change over a reference sample. For time course data fold-change was calculated in relation to the earliest stages (E7.75 *in vivo*, D0 ESC models), whilst for drug treated experiments fold change was calculated relative to control (DMSO) samples. Non-template controls were performed by replacing cDNA with water, to test for non-specific amplification.

## Western blot analysis

Protein was extracted on ice using direct lysis of cells with NP-40 extraction buffer (150 mM NaCl, 1.0% NP-40, 50 mM Tris (pH8.0)) and 1x protease and phosphatase inhibitor cocktail (Roche, England). Lysates were span at 10,000 rpm for 20 min at 4°C and supernatant collected. An aliquot was taken for protein quantification using a DC protein assay (Bio-rad, England). Supernatants were prepared for SDS-PAGE with the addition of 4x Laemmli sample buffer and boiling at 95°C for 5 min. Western blotting was performed using standard SDS-PAGE methods using HRP-conjugated secondary antibodies and enhanced chemiluminescence detection (GE Healthcare, England).

## Statistics

All data involving beat rate and qPCR gene levels was compared using one-way ANOVA followed by a Tukey test for multiple comparisons. In cases where the raw data failed to map to a normal distribution with consistent variance, we applied Taylor's law to choose the best transformation for the data. All data to be analysed passed the Shapiro-Wilk normality test and Bartlett test for homogeneous variances. To compare the number of affected embryos upon acute treatments, a Freeman-Halton extension of Fisher exact probability test was applied due to a smaller number of samples. To compare the effect of different treatments on the number of beating EBs, due to a large number of samples, a Chi Square test with a Bonferroni correction for multiple comparisons was performed instead. Principal Component Analysis (PCA) was carried out to directly compare temporal gene expression data from EBs with that derived from embryos; each biological replicate for embryonic stage was composed of 5–6 embryos and for EBs was composed of 10–80 EBs, depending on the day of differentiation. The data from each sample was normalized by assessing the ratio with the maximum value of all samples. A log transformation was applied to the normalized data and the principal components were calculated using R environment. The 3D representation of the PCA was constructed by plotting the 3 first components given that these explained more that 95% of the observed variance. A hierarchical clustering analysis was performed using Ward's minimum variance method as a more precise clustering algorithm to divide the samples in different groups (*Ward, 1963*).

## Image analysis

A variant of absolute image filter was used to visualize and plot measurements of cardiac contractions in the developing cardiac crescent as described elsewhere (*Chen et al., 2014*). Briefly, pixel displacement, indicative of contractions, was visualized and represented by increased grey levels within the crescent. Change in pixel intensity was assessed in a selected region, to reveal the contraction dynamics. Background $Ca^{2+}$ signal was subtracted from all frames of a given time-lapse using ImageJ. To obtain the profiles for $Ca^{2+}$ transients, regions of interest were plotted using the ratio between observed fluorescent and minimum fluorescent (F/F0) after background subtraction.

## Acknowledgements

This work was generously supported by the British Heart Foundation (RCVT, AMAM, SMD; CH/11/1/28798 and RG/13/9/303269 to PRR) including core support from the BHF Oxbridge Regenerative Medicine Centre (RM/13/3/30159), the BBSRC (REI BB/F011512/1 to SS) and The Wellcome Trust (C-mC; WTSIA 103788/Z/14/Z and 105031/C/14/Z to SS). We thank Liz Robertson for provision of the Eomes-GFP ESC line and Sean Wu for the Nkx2.5-eGFP ESC line.

## Additional information

### Funding

| Funder | Grant reference number | Author |
|---|---|---|
| British Heart Foundation | 4-year DPhil programme FS/12/69/3008 | Richard CV Tyser<br>Antonio MA Miranda |
| British Heart Foundation | BHF Oxbridge Regenerative Medicine Centre RM/13/3/30159 | Shankar Srinivas<br>Paul R Riley |
| Biotechnology and Biological Sciences Research Council | REI BB/F011512/1 | Shankar Srinivas |
| The Wellcome Trust | WTSIA 103788/Z/14/Z | Shankar Srinivas |
| The Wellcome Trust | 105031/C/14/Z | Shankar Srinivas |
| British Heart Foundation | CH/11/1/28798 | Paul R Riley |
| British Heart Foundation | RG/13/9/303269 | Paul R Riley |

The funders had no role in study design, data collection and interpretation, or the decision to submit the work for publication.

## Author contributions

RCVT, AMAM, Acquisition of data, Analysis and interpretation of data, Drafting or revising the article; C-mC, Acquisition of data, Analysis and interpretation of data, Contributed unpublished essential data or reagents; SMD, SS, PRR, Conception and design, Analysis and interpretation of data, Drafting or revising the article

## Author ORCIDs

Chiann-mun Chen, http://orcid.org/0000-0002-7007-3649
Paul R Riley, http://orcid.org/0000-0002-9862-7332

## Ethics

Animal experimentation: All animal experiments were carried out according to UK Home Office project license PPL 30/3155 Compliant with the UK Animals (Scientific Procedures) Act 1986.

## Additional files

### Supplementary files

• Supplementary file 1. Embryo staging specific to the early developing heart from cardiac crescent to linear heart tube. (a) Morphological criteria of stages of cardiac crescent development. Different stages of cardiac crescent development (stage 0 to stage 3) were defined based on the ratio between the weight and maximum width of the cardiac crescent. As development progresses the width decreases and the maximum height increases. These stages are a more accurate representation of developmental stage than more widely used staging criteria such as somite number and embryonic day. (b) Statistical comparison of qRT-PCR results of isolated cardiac crescents. Table of p-values obtained from ANOVA and a post-hoc Tukey test on the qRT-PCR results obtained from isolated cardiac crescents, comparing whole embryos pre-cardiac crescent formation (E7.75), at all stages of cardiac crescent development (stage 0 to stage 3) and linear heart tube (LHT) stage. (c) Statistical comparison of qRT-PCR results of isolated head folds. Table of p-values obtained from ANOVA and a post-hoc Tukey test on the qRT-PCR results obtained from isolated head folds, comparing whole embryos pre-cardiac crescent formation (E7.75), at all stages of cardiac crescent development (stage 0 to stage 3) and linear heart tube (LHT) stage. (d) Analysis from SACO inhibition experiments at Stage 0. Table of results detailing individual inhibitor experiments carried out on SACOs at stage 0. Information includes embryo ID, inhibitor used, absolute number of SACOs observed, before and after treatment, ratio of SACOs maintained after treatment, percentage inhibition, area containing SACOs and length of imaging. (e) List of antibodies. List of primary antibodies used for immunostaining and western blot analyses and secondary antibodies used for all the experiments with source and dilution used. (f) List of primers for qRT-PCR. List of primer pairs used for qRT-PCR analysis and respective amplicon size.

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
