## [Decision Letter]

[Editors’ note: a previous version of this study was rejected after peer review, but the authors submitted for reconsideration. The previous decision letter after peer review is shown below.]

Thank you for choosing to send your work entitled "Calcium handling precedes cardiac differentiation to initiate the first heart beat" for consideration at *eLife*. Your full submission has been evaluated by Janet Rossant (Senior editor) and three peer reviewers, as well as a member of our Board of Reviewing Editors, and the decision was reached after discussions between the reviewers. Based on our discussions and the individual reviews below, we regret to inform you that your work will not be considered further for publication in *eLife*.

We appreciate the importance of this study on the establishment of calcium gradients and its implications for cardiogenesis. We also appreciate the technical advances in this paper which presents a multidisciplinary approach and includes functional studies. However, a number of points raised by the reviewers indicate problems with the paper as it stands. Notably new mechanistic insights are weak. There is a lack of data on LTCC Cav1.3. The mechanistic link between early NCX-1 dependent asyncronous activity and maturation of electrical activity and excitation contraction coupling is lacking – blocking experiments in embryo culture are inadequate. There is also a question about novelty. The Introduction is weak and does not adequately cover current available papers on the subject. Reconsideration of experimental approaches in the context of earlier studies should lead to the design of further experiments such as genetic ablation of key components or electrical measurements in embryo culture.

We consider that the issues raised by the reviewers cannot be addressed in the short time frame required by *eLife* for return of a revised manuscript. However, we would encourage you carry out further experiments and if you then feel that you have addressed all the issues, *eLife* remains open to receiving a revised manuscript in the future for consideration as a new submission.

*Reviewer #1:*

The authors present an interesting study demonstrating calcium-mediated contractile activity in the mouse cardiac crescent earlier than previously thought. The study is strengthened by the use of both embryonic data and ESC-derived cardiomyocyte data. The experiments follow a logical progression with the data and figures presented clearly. I recommend accepting the manuscript with revision and additional experiments.

While concise scientific communication is appreciated, the Introduction is a bit too concise (one paragraph composed of three sentences). The manuscript would benefit from additional background and justification for the current study.

Given the strength of the assertion that this study is a definitive analysis of early crescent contractility, the sarcomere immunostaining should be more thorough and expanded to include other sarcomeric proteins that spatiotemporally indicate sarcomere formation. Although the current data is supportive of the conclusions presented, cTnT alone is not sufficient in this regard. At minimum, immunostaining for α-actinin, a Z-disc protein would much improve and strengthen this data. Further, inclusion of staining for myomesin, an M-band protein would be much more definitive especially if double stained showing alternating actinin/myomesin.

The gene expression studies should be complemented by whole mount in situ hybridization. This is especially important for Figure 4 as neither Slc8a1 or Cacna1c are solely restricted to the heart and the gene expression was performed on whole embryos.

More information is needed to evaluate the gene expression studies. At minimum, primer details, amplicon details, replicates (both biological and technical) and PCR parameters should be reported. In the absence of validating the reference genes for qPCR, what is the rationale for choosing HPRT for in vivo samples and GAPDH for ESCs? How is fold change calculated? Is it in relation to E7.5 for embryos and D0 for ESCs? The following references should be helpful in this regard.

Johnson G, et al. Methods Mol Biol. 2014;1160:5-17.

Bustin SA, et al.. Nat Methods. 2013 Nov;10(11):1063-7.

Bustin SA, et al. BMC Mol Biol. 2010 Sep 21;11:74.

*Reviewer #2:*

The manuscript addresses the earliest stages of electrical activity in the mammalian heart and ES cell-derived EBs and in particular the relative requirements of the Na/Ca exchanger NCX1 and L-type Ca channel (LTCC). The main implications of the work are when Ca transients are established and how this relates to cardiogenesis. In general, the work is well written and the data interesting and relevant to the above issues. The functional and multidisciplinary approach to cardiac development is a strength. For the most part the experiments are carefully performed and results make a strong contribution.

1) The novelty of the findings are diminished somewhat by prior publications on genetic deletion of NCX and LTCC genes in mice and fish. The novelty of this study lies in the detailed assessment of beating, Ca transients, and relative requirements for NCX1 and LTCCs. I recommend that some of the material relegated to the Discussion come forward to the Introduction to make clear the state of the field including the broad findings relating to NCX1 and LTCC pharmacological and genetic inhibition in the context of embryogenesis, and the novelty of the present findings in a summary paragraph. Background studies should cover what has been observed in the chick embryo wrt beating and pacemaker function, and electrophysiology.

2) The text is not rigorous in defining the dose of drugs used and the duration of use in each experiment. Different doses and treatments are mentioned. In addressing the relative requirements for NCX1 and LTCCs, dose response data would seem important as different drugs may have different Ki etc. It seems important to address this formally throughout or to do specific experiments that incorporate dose responses.

3) Furthermore, rationale is not given for differing doses between experiments e.g. for CB-DMB is used at 20 μm in experiments shown in Figure 5 but at 1mM in those in Figure 6A-D. This is a massive difference. The authors must provide full rationale for these choices in light of the issue raised under point 2.

4) The experimental design in the experiment described at the end of page 8 onwards does not allow the stated conclusions to be drawn. First, duration of drug treatment is unknown. Second, the design of the sequential experiment is informative but does not allow the authors to conclude that the earlier NCX-1-dependent waves are essential for subsequent differentiation and beating.

5) In the last paragraph of the subsection “NCX1 is essential for the initiation of Ca^2^ transients in the developing cardiac crescent”. the effects on GFP from Nkx2-5GFP should be quantified, preferably by FACS.

6. In the last paragraph of the subsection “NCX1 is essential for the initiation of Ca^2^ transients in the developing cardiac crescent”. This is a key experiment that addressed an important theme for the paper; the response of the cardiac network to inhibition of NCX1 and LTCC was poorly analysed (only 4 genes analysed, two of these encoding the channels, and only at one time point). Effects could be secondary.

7) In the subsection “Inward Ca^2^ via NCX1 and downstream CAMKII signalling promotes cardiomyocyte differentiation”; inhibition of beating in low Ca is not surprising. How do the results show that Ca is required for maturation and differentiation?

8) In the subsection “NCX1 is essential for cardiac crescent formation during development”. For such an important point, actin staining of embryos treated with inhibitors and PCR are insufficient to draw the broad conclusion that there is a lack of crescent cells. This experiment needs far more development. What happens to the pre-cardiac cells and cardiac progenitors over time needs to be analysed with appropriate markers?

*Reviewer #3:*

The manuscript "Calcium handling precedes cardiac differentiation to initiate the first heart beat" by Tyser et al. tries to address the importance of calcium cycling for cardiac crescent formation in the embryonic heart by Ca^2^ imaging, gene expression analysis and pharmacological blocking during culture of mouse embryos and embryonic stem cells. The main result is that the sodium-calcium-exchanger (NCX) is important for cardiomyocyte development and function because pharmacological block of NCX reduces Ca^2^ transients and beating as well as lowers expression of cardiac genes and delays cardiac crescent formation.

Although the manuscript presents major technical advancements in this field including mouse embryo culture and recording of Ca^2^ transients from the cardiac crescent prior to beating, the study remains descriptional and the proposed mechanism is based on over interpretation of results and assumptions including drug specificity of long term drug application.

Most importantly the authors have not measured or discussed the involvement of voltage signals in cardiomyocytes and several other known facts on the physiology of embryonic heart cells or on the pacemaking mechanism in the embryonic heart.

The authors write "NCX1, have not previously been implicated in the initiation event of cardiac contraction, nor investigated at the earliest stages of heart development coincident with the onset of beating" and in the Abstract "how the initial contractions are established have not been described". This is incorrect. It is well known that in the embryonic heart cell, a periodic intracellular calcium oscillation (IP3-dependent) is driving pacemaking and translated into voltage fluctuations specifically by the NCX (forward mode of the electrogenic pump). These fluctuations can induce action potentials that synchronize the cardiac tissue and therefore NCX is essential for cardiac development (please see for instance: Excitation-Contraction Coupling of the Mouse Embryonic Cardiomyocyte" Rapila R et al. 2008 and the other papers by the Tavi group or "Intracellular Ca^2^ Oscillations, a Potential Pacemaking Mechanism in Early Embryonic Heart Cells" Sasse P et al. 2007).

Furthermore, it is well known that intact Ca^2^ cycling is important for cardiac gene expression and heart development. The authors should clearly cite the previous work and specifically describe the novelty and mechanism of their study.

The main conclusion of the blocker experiments is that NCX reverse mode is important for Ca^2^ entry and gene regulation but how this should induce periodic beating or pacemaking has not been addressed or discussed. NCX and can only work in reverse mode when intracellular Na concentration is excessively high during an action potential (although controversial if this really happens, or if the effect is only a less effective forward mode). This has not been shown for embryonic cardiomyocytes and is highly unlikely because of the lack of Na channels or Na driven action potentials at this stage.

The (confusing) idea of a reverse mode NCX comes from the literature that state that at low dosages (EC50 in the low µM range) KB-R is preferential blocking a reverse mode. Although oocyte experiments with non-physiological intracellular Na concentrations suggest this, other experiments on cardiomyocytes (e.g. "Direction-independent block of bi-directional Na /Ca^2^ exchange current by KB-R7943 in guinea-pig cardiac myocytes" by Kimura, J et al. 1999) show that this specificity is not existent and other papers show that this is highly depending on intracellular Na concentrations and drug dosage used. Thus without electrophysiological analysis, it remains unclear, if the inhibitory effect on beating and cardiac crescent formation after block of NCX by KB-R or CB-DMB is due to effects on of Ca^2^ oscillations, pacemaking, translemmal Ca^2^ entry or exit, membrane potential fluctuation, synchronization by action potentials, imbalance of Ca^2^ homeostasis or off-target effects that occur especially when applied long term.

In summary, the presented data, although in part novel and interesting, together with the known literature on embryonic cardiomyocytes and on the physiology of ECC, LTCC and NCX function does not allow to conclude a mechanism.

Specific major points:

The Introduction is too short and only very general. Specifically, several known facts on initiation of the first heart beat and the role of calcium gene regulation during embryonic heart development should be cited. For instance, several earlier publications have tried to address "the important question of when and how contractile activity of cardiomyocytes is first initiated during development" (see above). In addition to cite these earlier work in introduction it should be better pointed out, why these earlier studies are not sufficient and the current study is required (which is easy, because of the nice in vivo data).

Also the first paragraph of the section "A role for the Sodium-Calcium exchanger…" belongs to the Introduction. Please describe the well-known ECC in the adult heart a bit less but introduce more the known facts on ECC and the importance of NCX for pacemaking in the embryonic heart (see above).

Because of the low resolution of the supplied PDF file, the cardiac maturation and cross striation at the different crescent sages is difficult to review. Maybe providing inserts to highlight the specific patterns (especially stage 1 2) would help. Is cross striation completely absent in stage 0?

The finding of Ca^2^ transients at stage 0 is fascinating. Maybe better call them "spontaneous asynchronous Ca^2^ oscillations" also to highlight the difference semantically to the (action potential-based?) coupled transients at stage 1 . What is the average frequency and is there a correlation between frequency and TTP (maybe more interesting to show than TTP vs TT_1/2_M which lacks a conclusion)? Is there evidence that transients are coupled between individual cells (from the video this seems to be sometimes the case). Although not in the scope of the authors´ manuscript, recording of membrane potential is essential to understand the nature of cell-cell coupling of these fascinating data.

Figure 3: Normalizing gene expression data to D0 or E7.5 (where most genes are not expressed) make less sense, adds noise and masks information. Please consider normalization to the time point of the highest value.

Unfortunately, Cacna1d (Cav1.3), which is the dominant isoform of LTCC in the early embryo (see "Subtype switching of L-Type Ca 2 channel from Cav1.3 to Cav1.2 in embryonic murine ventricle". Takemura H et al. 2005) was not analyzed and therefore the expression data of Cacna1c does not suggest a minor role of LTCC, but just highlights the well-known isoform switch (see also "Functional Embryonic Cardiomyocytes after Disruption of the L-type α1C (Cav1.2) Calcium Channel Gene in the Mouse"). The analysis of other LTCC genes or western blot at different stages from heart tissue might help in this regard.

The establishment of an embryo culture for investigation of cardiac crescent or heart tube formation as well as for gene expression analysis is a definite plus and a main methodological advancement of this paper. Very nice! However, the fact that "Embryos cultured in CB-DMB were delayed in terms of cardiac crescent formation" is not supported by the one actin image shown (Figure 6O) although it is one of the main messages of the paper. Repeating of experiments, extensive histological analysis by specific crescent staining, and quantitation is suggested. Also a time course analysis of crescent formation with and without CB-DMB and KBR would enhance the paper. How many hours are the cardiac crescent forming delayed under NCX blockage?

Was the data in Figure 5 obtained from Ca imaging or from video microscopy? If here video microscopy was used, also data from Ca imaging should be added to see the acute (>1-2 minutes) effects of the three blockers on Ca^2^ transients and remaining Ca^2^ oscillations (before all Ca^2^ has left the cell and the impaired Ca^2^ homeostasis is blocking pacemaking).

In the Discussion the authors have claimed to use "two independent pharmacological channel blockers CB-DMB and KB-R7943, in embryo cultures", but only show CB-DMB data in Figure 6. Please also show the KB-R data on embryonic cultures.

The authors describe “sequential addition of CB-DMB to nifedipine-treated embryos after 15 mins, subsequently blocked the transients that were refractory to LTCC inhibition (not shown)." Please provide the data and statistics and explain the mechanism. What is the percentage of transients that were not blocked by LTCC inhibition? Was this observed at all stages or is this stage specific?

[Editors’ note: what now follows is the decision letter after the authors submitted for further consideration.]

Thank you for submitting your article "Calcium handling precedes cardiac differentiation to initiate the first heartbeat" for consideration by *eLife*. Your article has been reviewed by three peer reviewers, and the evaluation has been overseen by a Reviewing Editor and Janet Rossant as the Senior Editor. The reviewers have opted to remain anonymous.

The reviewers have discussed the reviews with one another and the Reviewing Editor has drafted this decision to help you prepare a revised submission.

The manuscript resubmitted after revision is much improved and is now under review with a view to publication of these interesting findings. The authors show that the sodium calcium exchanger (NCX) early, at what they define as stage 0, is essential for generating spontaneous asynchronous Ca^2^ oscillations (SACO) in the cardiac crescent and that these are essential for cardiac gene expression and subsequent development. However, interpretation of the results and the proposal that reverse mode NCX, allowing Ca^2^ to enter the cell, is involved in the mechanism are still problematic.

Firstly, in the experiments with inhibitors, the stage 0 experiments are critical. The authors use nifedipine to block voltage dependent Ca^2^ channels (VDCC), which interestingly at stage 0 has less effect on SACO, in contrast to later stages when it blocks activity. CB-DMB, which blocks NCX, affects SACO at stage 0, but it also blocks Ca^2^ exit mode. In the manuscript KB-R7943 is claimed to specifically block reverse mode NCX but it was not used at stage 0. This result should be shown. The finding that this inhibitor blocks activity at stages 1 and 2 is not so important because nifedipine does this too. In their interpretation the authors should be aware of controversy about the action of KB-R7943.

The SACO shown in Figure 2 and Video 3 is one of the most important pieces of data and in fact is the major novelty. Please provide more data here including n-numbers, statistics and report reproducibility of these findings. Are SACO observed in every stage 0 embryo? What is the Ca^2^ cycling frequency and amplitude and how often are single uncoupled (Figure 2´) and multiple coupled cells (Figure 2´´) observed?

The experiments on blocking SACO with CB-DMB in contrast to nifedipine lead to one of the most important findings, however the analysis and statistics are very limited. Please provide absolute numbers of cells with SACO and original data (example traces, videos) to enable the reader to follow the analysis and statistics. Line 310: What does "0.87 0.083, mean SEM" refer to? How are cells without SACO identified if they do not show Ca^2^ signals? How is the variance of cells with SACO per embryo? ANOVA comparisons are required to take these variations in control embryos into account.

The novel Figure 3—figure supplement 1 on RyR/APD treatment is confusing as it does not follow the authors (very good) stage classification (stage 0, 1, 2, 3, LHT) but has a novel classification (turned/pre-turned embryo). Please label data as stage 0, stage 1 and LHT, at least. Please add statistics, error bars and original Ca^2^ imaging traces. A DMSO control for stage 3 is missing.

Figure 4: the error bars are missing. In fact, it is not even fully clear from the methods what% inhibition means. Please discriminate between complete block of beating (movement in DIC images), reduction of frequency, complete block of Ca^2^ transients and reduction of Ca^2^ transient amplitude. Also please provide time courses and report the delay of drug action (should be easy as Ca^2^ imaging was performed at baseline and at 5, 15 and 30 minutes post drug treatment).

In general, in the text it is important to clarify what is meant by inhibition and partial inhibition.

Although the pharmacological block using CB-DMB shown in Figure 4 lets the authors conclude that Ca^2^ entry though NCX is the basis for the SACO, the mechanism of how this could be periodic and lead to oscillations in Ca^2^ is unclear. The authors are correct that NCX can (under special circumstances such as elevated intracellular Na or depolarized membrane potential) lead to Ca^2^ entry, but once this occurs, the electrogenic nature of NCX (~3 Na for 1 Ca^2^ ) will again hyperpolarize the cell and export Na until the equilibrium potential is reached and Ca^2^ flux ceases. Thus it is not possible that the NCX can generate periodic Ca^2^ oscillations alone. Reverse mode only occurs if periodic action potentials are depolarizing the cell but then cells should be synchronized (no SACO). Maybe Na levels are cycling or the Na/K-ATPase is involved? This could be clarified by testing whether SACO at stage 0 are blocked by RyR/2-APB or by SERCA blockers (Thapsigargin/CPA). These mechanistic considerations should be taken into account in the authors' Discussion.

On the mechanistic front, as it stands the paper does not permit definitive conclusions; the authors must be less dogmatic about the SACO/Ca^2^ cycling mechanism and introduce notes of caution in their Discussion. The authors are encouraged to attempt electrophysiological analysis since without this it remains unclear if the inhibitory effect on beating and cardiac crescent formation of NCX by KB-R7945 or CB-DMB is due to effects on SACO, translemmal NCX reverse mode Ca^2^ entry (mechanism unclear) or NCX forward mode Ca^2^ exit (NCX block-based Ca^2^ overload can also inhibit gene expression), imbalance of Ca^2^ homeostasis or off-target effects. These analyses as well as straight forward siRNA experiments, proposed in the first reviews prior to resubmission, should be feasible with the elegant embryo culture technology set up by the authors.

A second aspect which requires further revision is the gene expression data. This should be normalised relative to housekeeping control genes. Normalising to E7.75 when most genes are not expressed and there is mainly noise is not informative, preventing meaningful comparison between genes. For instance, in the current presentation the time course of NCX and Cav1.3 seems to be superimposable. The data should be represented as dCT (normalised only to housekeeping genes) and then the meaningful differences in time course/relative levels discussed.

[Editors' note: further revisions were requested prior to acceptance, as described below.]

Thank you for resubmitting your work entitled "Calcium handling precedes cardiac differentiation to initiate the first heartbeat" for further consideration at *eLife*.

Before publication, the following remaining concerns of one reviewer will need to be addressed:

1) Please provide information on the frequency, duration and regularity of SACO at stage 0 as requested and as stated in the subsection “The onset of Ca^2^ -handling in the cardiac crescent” ("variable frequencies and durations"). From the authors comment in the rebuttal and from Video 4 (very nice) it is well understandable that precise numbers are difficult to give but at lease state something like "Compared to Ca^2^ transients at later stages (Figure 2) the SACO were rare. During a ~20 s recording period we observed only 10.3 -x individual SACO per embryo (n=24) occurring in different sites. Consecutive SACO in a same site were rarely observed within the 20 s imaging windows and therefore we conclude that SACO in individual cells occur at a frequency < 3 bpm."

2) Please describe the meaning of Figure 2—figure supplement 1 with something like "During a SACO, Ca^2^ rises slowly and reaches the peak between 0.5 and 10 s. Cells with slow Ca^2^ influx also showed similar slow Ca^2^ efflux (Figure 2—figure supplement 1)."

3) Because of the very slow nature of SACO, also oscillations in energy production and ATP levels can be involved (in fact the most likely explanation of SR is not involved). Please add this to the Discussion (fourth paragraph).

4) The authors state that Ryanodine 2-APB only affected embryos that had already undergone cardiac looping but the earliest stage tested was stage 3. Although requested, the effect of Ryanodine 2-APB on SACO was not analyzed. Thus it seems to be fair to include in the revised Discussion on the SACO mechanism (fourth paragraph) something like: "Because we have not tested the involvement of SR function at stage 0, we cannot fully exclude that periodic Ca^2^ releases from the SR are involved in the generation of SACO as reported before as a pacemaking mechanism in early embryonic cardiomyocytes (cite adequate references).

---

## [Author Response]

[Editors’ note: the author responses to the previous round of peer review follow.]

*We appreciate the importance of this study on the establishment of calcium gradients and its implications for cardiogenesis. We also appreciate the technical advances in this paper which presents a multidisciplinary approach and includes functional studies. However, a number of points raised by the reviewers indicate problems with the paper as it stands. Notably new mechanistic insights are weak.*

Regarding the editorial summary on mechanistic insight, our study is the first to identify the onset of cardiac function and physiology within the embryo at significantly earlier stages of heart development (during formation of the cardiac crescent) than previously described. Prior studies have focused on linear heart tube looping (E8.5) stages and have relied extensively on *in vitro* analyses of isolated cells (Sasse et al. 2007; Rapila et al., 2008). In revision, we have now added additional experiments using the inhibitors Ryanodine and 2-APB to target the sarcoplasmic reticulum (SR) as a source of Ca^2^ and whilst we found these inhibited cardiac function at later stages of heart development (in turned embryos at ~E8.5), there was no effect at the earliest onset of Ca^2^ handling and contractile function. This new data suggests that mechanistically the SR is not required for the initiation of spontaneous asynchronous calcium oscillations (SACOs) or Ca^2^ transients within the cardiac crescent. A requirement for SR function develops at later stages and further supports the role of NCX in initiating the earliest Ca^2^ influx and the generation of SACOs.

In this study we present the following:

We attribute a mechanism for the onset of Ca2 handling and cardiac function at significantly earlier stages of heart development than previously appreciated;

Our data from culture and imaging of live embryos is representative of the in vivo setting and avoids caveats associated with isolated cell preparations and *in vitro* studies;

We exclude an essential role for the SR and reveal that sarcolemmal Ca2

flux via NCX, and then sequentially LTCCs, predominates within the early developing heart as distinct from adult modes of Ca2 -handling;

NCX alone is required for the generation of SACOs prior to the initiation of contraction.

*There is a lack of data on LTCC Cav1.3.*

We have now included detailed expression profiling of the LTCC subunit Cav1.3, in the revised manuscript. However, whilst this is of interest, our use of Nifedipine blocks all LTCC isoforms, inhibiting both Cav1.2 and Cav1.3 and therefore, is a viable reagent to assess a general role for LTCCs in early Ca^2^ handling. Application of Nifedipine at stage 0, did not inhibit SACOs in the forming cardiac crescent, excluding an early role for both Cav1.2 and Cav1.3.

*The mechanistic link between early NCX-1 dependent asyncronous activity and maturation of electrical activity and excitation contraction coupling is lacking – blocking experiments in embryo culture are inadequate. There is also a question about novelty.*

Whilst we are interested in understanding how SACOs mature into propagating synchronized transients, this was not the focus in this study and beyond the scope of the current manuscript. The focus on the study was detailing how cardiac function is first initiated and the downstream consequences of this in terms of how Ca^2^ -handling and contractile function feed back onto cardiomyocyte differentiation/maturation and early cardiac morphogenesis functioning.

*The Introduction is weak and does not adequately cover current available papers on the subject. Reconsideration of experimental approaches in the context of earlier studies should lead to the design of further experiments such as genetic ablation of key components or electrical measurements in embryo culture.*

The Introduction has been re-written to add more detail, citing the existing studies on the development of cardiac function and more comprehensively addressing those that have established the current dogma of contraction occurring at the linear heart tube stage.

*Reviewer #1:*

*The authors present an interesting study demonstrating calcium-mediated contractile activity in the mouse cardiac crescent earlier than previously thought. The study is strengthened by the use of both embryonic data and ESC-derived cardiomyocyte data. The experiments follow a logical progression with the data and figures presented clearly. I recommend accepting the manuscript with revision and additional experiments.*

We thank the reviewer for the positive comments and suggestion to accept the manuscript with revision and additional experiments. We have now addressed the major points below with further detailed experimental analyses and significant new data.

*While concise scientific communication is appreciated, the Introduction is a bit too concise (one paragraph composed of three sentences). The manuscript would benefit from additional background and justification for the current study.*

We have now expanded the Introduction to cover more background and justification for this study.

*Given the strength of the assertion that this study is a definitive analysis of early crescent contractility, the sarcomere immunostaining should be more thorough and expanded to include other sarcomeric proteins that spatiotemporally indicate sarcomere formation. Although the current data is supportive of the conclusions presented, cTnT alone is not sufficient in this regard. At minimum, immunostaining for α-actinin, a Z-disc protein would much improve and strengthen this data. Further, inclusion of staining for myomesin, an M-band protein would be much more definitive especially if double stained showing alternating actinin/myomesin.*

We thank the reviewer for the suggestions and have now included a developmental timecourse of sarcomeric α-actinin and Myomesin immunostaining, which has allowed us to more conclusively define sarcomere formation (see revised Figure 1; subsection “Staging of early cardiac development and sarcomeric assembly”, second paragraph). In addition, we have performed qRT-PCR on the genes encoding cardiac troponin T, myomesin and sarcomeric α-actinin, which all revealed significant up-regulation in isolated cardiac crescents between stage 0 and stage 1 (See revised Figure 1; and the aforementioned paragraph).

*The gene expression studies should be complemented by whole mount in situ hybridization. This is especially important for Figure 4 as neither Slc8a1 or Cacna1c are solely restricted to the heart and the gene expression was performed on whole embryos.*

We have expanded our data to now include gene expression from both isolated cardiac crescents and head folds to address the issue of *Slc8a1* or *Cacna1c* being expressed both in heart and extra-cardiac lineages. By isolating tissues from within the embryo and performing qRT-PCR we are able to more accurately quantify gene expression, as well as assess multiple genes per time point, which is why we have retained a qRT-PCR based-approach as opposed to performing whole mount in situhybridisation. Whilst *Slc8a1* and *Cacna1c* both increased in the heart samples there were no changes in expression within the head folds (see Figure 3—figure supplement 1; subsection “Ion channels are expressed during the earliest stages of heart development”) confirming our original data. In addition, our immunostaining analyses for NCX and LTCC provides spatial localisation on protein expression, negating the need to perform whole mount in situ hybridisation.

*More information is needed to evaluate the gene expression studies. At minimum, primer details, amplicon details, replicates (both biological and technical) and PCR parameters should be reported. In the absence of validating the reference genes for qPCR, what is the rationale for choosing HPRT for* in vivo *samples and GAPDH for ESCs? How is fold change calculated? Is it in relation to E7.5 for embryos and D0 for ESCs? The following references should be helpful in this regard.*

Johnson G, et al. Methods Mol Biol. 2014;1160:5-17.

Bustin SA, et al.. Nat Methods. 2013 Nov;10(11):1063-7.

*Bustin SA, et al. BMC Mol Biol. 2010 Sep 21;11:74.*

We have now included further details to more comprehensively evaluate our gene expression studies (see revised [Supplementary-material SD1-data]; subsection “RNA-extraction and qPCR”). [Supplementary-material SD1-data] includes a list of primers used as well as sequence information, amplicon size, primer source, PCR parameters and further details are now also included in the revised Methods section. All samples were run in triplicate to account for technical variability and figure legends now state the number of biological replicates. Reference genes were initially validated by qPCR and this was the rationale for not using HPRT as a housekeeping gene when assessing gene expression during ESC differentiation, given the HPRT Ct values decreased during differentiation, as previously reported (Murphy and Polak, 2002). We have now used multiple housekeeping genes for each experiment, to provide us with more comprehensive reference expression levels, as now stated in the revised methods. For time course data fold-change was calculated in relation to the earliest stages (E7.75 in vivo, D0 ESC models), whilst for drug treated experiments fold change was calculated relative to control (DMSO) samples; this is now included in the revised methods as requested.

*Reviewer #2:*

*The manuscript addresses the earliest stages of electrical activity in the mammalian heart and ES cell-derived EBs and in particular the relative requirements of the Na/Ca exchanger NCX1 and L-type Ca channel (LTCC). The main implications of the work are when Ca transients are established and how this relates to cardiogenesis. In general, the work is well written and the data interesting and relevant to the above issues. The functional and multidisciplinary approach to cardiac development is a strength. For the most part the experiments are carefully performed and results make a strong contribution.*

We thank the reviewer for their positive comments and for highlighting both the strength of using a multidisciplinary approach to address cardiac development as well as the view that these results make a strong contribution to the field.

*1) The novelty of the findings are diminished somewhat by prior publications on genetic deletion of NCX and LTCC genes in mice and fish. The novelty of this study lies in the detailed assessment of beating, Ca transients, and relative requirements for NCX1 and LTCCs. I recommend that some of the material relegated to the Discussion come forward to the Introduction to make clear the state of the field including the broad findings relating to NCX1 and LTCC pharmacological and genetic inhibition in the context of embryogenesis, and the novelty of the present findings in a summary paragraph. Background studies should cover what has been observed in the chick embryo wrt beating and pacemaker function, and electrophysiology.*

We thank the reviewer for highlighting the novelty of our manuscript and have rewritten the introduction accordingly. In terms of this paper’s findings being diminished by prior publications on genetic deletion models in mice and fish, it is important to note that genetic loss of function for NCX1 has produced a wide array of embryonic heart phenotypes, despite the fact that all the published studies targeted the same exon of NCX1. The hearts of the various NCX1 mutants range from arresting at the linear heart tube stage to completed looping morphogenesis and there is no insight into a putative early role in establishing Ca^2^ -handing and contraction given the varied lethality. There is also insufficient evidence in these prior studies to assess whether complete KO has occurred and/or whether compensation is occurring via NCX1 splice variants, other NCX isoforms or alternative mechanisms of Ca^2^ influx. By using pharmacological inhibition (with two independent specific NCX-inhibitors) at defined stages of development, we are able to rule out any compensatory effects arising from germ-line genetic ablation and moreover, simultaneously target multiple isoforms to converge on a role for NCX during the onset of cardiac function.

*2) The text is not rigorous in defining the dose of drugs used and the duration of use in each experiment. Different doses and treatments are mentioned. In addressing the relative requirements for NCX1 and LTCCs, dose response data would seem important as different drugs may have different Ki etc. It seems important to address this formally throughout or to do specific experiments that incorporate dose responses.*

We appreciate the problems with using pharmacological inhibition and hope that the revised manuscript is more thorough in defining the doses used. Drugs were selected based on previous publications which also defined the optimal working concentrations as effective when applied to *in vitro* cell-based assays. We tended to use slightly higher concentrations, albeit within the same µm range, due to penetration issues when using whole embryos. It is reassuring to note that whilst NCX inhibitors blocked transients at early stages, the effect was lost at later stages, thus excluding non-specific effects such as impaired cell survival, etc.

*3) Furthermore, rationale is not given for differing doses between experiments e.g. for CB-DMB is used at 20 μm in experiments shown in Figure 5 but at 1mM in those in Figure 6A-D. This is a massive difference. The authors must provide full rationale for these choices in light of the issue raised under point 2.*

We apologise for the confusion; 1mM was an error, which has now been corrected. The highest concentration of CB-DMB used was 20 µM in the acute inhibitor experiments. The reason why higher concentrations were used when working with whole embryos is due to the penetration issues, associated with performing in vivo embryo Ca^2^ measurements. Unfortunately, we are unable to definitively measure the concentration of drug to which the tissue was actually exposed.

*4) The experimental design in the experiment described at the end of page 8 onwards does not allow the stated conclusions to be drawn. First, duration of drug treatment is unknown. Second, the design of the sequential experiment is informative but does not allow the authors to conclude that the earlier NCX-1-dependent waves are essential for subsequent differentiation and beating.*

We thank the reviewer for pointing this out and have included new experiments in revised Figure 4 and reworded the accompanying text (subsection “Ca^2^ transients within the forming cardiac crescent are dependent upon NCX1”) to address this issue. From these new experiments we are able to definitively conclude that NCX1-dependent waves are required for the generation of SACOs which precedes LTCC function. Drug treatment was maintained for 15 mins and assessed at both 5 mins and 15 mins as now stated in the aforementioned subsection and the revised Methods.

*5) In the last paragraph of the subsection “NCX1 is essential for the initiation of Ca^2^ transients in the developing cardiac crescent”. the effects on GFP from Nkx2-5GFP should be quantified, preferably by FACS.*

We have now quantified Nkx2.5-GFP by assessing the percentage of GFP area across multiple images taken from different experiments (see Figure 5—figure supplement 3, subsection “Ca^2^ channel blockade inhibits ESC differentiation into cardiomyocytes”). We have carried out independent sets of experiments using two entirely distinct ESC lines and observed the same phenotypes.

*6. In the last paragraph of the subsection “NCX1 is essential for the initiation of Ca^2^ transients in the developing cardiac crescent”. This is a key experiment that addressed an important theme for the paper; the response of the cardiac network to inhibition of NCX1 and LTCC was poorly analysed (only 4 genes analysed, two of these encoding the channels, and only at one time point). Effects could be secondary.*

These experiments include four key cardiac genes, but also additional analysis of *Slc8a1, Cacna1c* and *CamKIId*. We initially focused on a single end point when control cardiomyocyte would have been expected to be at their most differentiated. In revision, we have now also included a time course of ESC differentiation with and without CB-DMB, to more carefully analyse the effect of inhibition. This new data reveals that inhibition of NCX occurs during the initiation of cardiomyocyte differentiation between day 4 and 7 and supports our previous conclusions (see Figure 5—figure supplement 1).

*7) In the subsection “Inward Ca^2^ via NCX1 and downstream CAMKII signalling promotes cardiomyocyte differentiation”; inhibition of beating in low Ca is not surprising. How do the results show that Ca is required for maturation and differentiation?*

Percentage of beating embryoid body analysis was performed in media containing 1.8 mM Ca^2^ , as stated in the Methods, therefore a reduced percentage of beating embryoid bodies would represent a decrease in cardiomyocyte formation and would not be just a physiological consequence of low Ca^2^ .

*8) In the subsection “NCX1 is essential for cardiac crescent formation during development”. For such an important point, actin staining of embryos treated with inhibitors and PCR are insufficient to draw the broad conclusion that there is a lack of crescent cells. This experiment needs far more development. What happens to the pre-cardiac cells and cardiac progenitors over time needs to be analysed with appropriate markers?*

We have now repeated these experiments for multiple embryos, using immunohistochemistry for the contractile protein cTnT as a marker of differentiation and the transcription factor Nkx2.5 as a marker of cardiac progenitors within the crescent (see Figure 5; subsection " NCX1 is essential for cardiomyocyte differentiation and cardiac crescent formation”). By performing this more in-depth analysis, we are confident in concluding that inhibition of NCX by CB-CMB leads to a decrease in mature cardiomyocytes which is not observed with inhibition of LTCCs by nifedipine treatment.

*Reviewer #3:*

*The manuscript "Calcium handling precedes cardiac differentiation to initiate the first heart beat" by Tyser et al. tries to address the importance of calcium cycling for cardiac crescent formation in the embryonic heart by Ca^2^ imaging, gene expression analysis and pharmacological blocking during culture of mouse embryos and embryonic stem cells. The main result is that the sodium-calcium-exchanger (NCX) is important for cardiomyocyte development and function because pharmacological block of NCX reduces Ca^2^ transients and beating as well as lowers expression of cardiac genes and delays cardiac crescent formation.*

*Although the manuscript presents major technical advancements in this field including mouse embryo culture and recording of Ca^2^ transients from the cardiac crescent prior to beating, the study remains descriptional and the proposed mechanism is based on over interpretation of results and assumptions including drug specificity of long term drug application.*

We are grateful for the positive comment regarding major technical advancements in this field including mouse embryo culture and recording of Ca^2^ transients from the cardiac crescent prior to beating. We have had to include detailed descriptions of what we observe at the very earliest stages of cardiac crescent formation, Ca^2^ handling and contractility given this has not been described before and is attributed to our advanced live imaging of mouse embryos. The pharmacological blockade of NCX1 and LTCCs provides mechanistic insight into the requirement for the exchanger, acting in reverse mode, to initiate Ca^2^ influx and the appearance of SACOs prior to full transients and contractility within the crescent. Any issue with drug specificity is significantly reduced by our use of two independent inhibitors for NCX and the fact that we observe consistent results across independent models of mouse embryos and ESC-cardiomyocyte differentiation.

*Most importantly the authors have not measured or discussed the involvement of voltage signals in cardiomyocytes and several other known facts on the physiology of embryonic heart cells or on the pacemaking mechanism in the embryonic heart.*

Pace-making appears not to apply to stages earlier than the looping heart tube, consistent with our focus on the initiation of Ca^2^ handling and asynchronous oscillations (without pacing) as precedes contractile function. We have not made any statements regarding voltage signals, as we have not developed the whole tissue approach to record them in early embryos and we respectfully suggest this is beyond the scope of the current manuscript.

*The authors write "NCX1, have not previously been implicated in the initiation event of cardiac contraction, nor investigated at the earliest stages of heart development coincident with the onset of beating" and in the Abstract "how the initial contractions are established have not been described". This is incorrect. It is well known that in the embryonic heart cell, a periodic intracellular calcium oscillation (IP3-dependent) is driving pacemaking and translated into voltage fluctuations specifically by the NCX (forward mode of the electrogenic pump). These fluctuations can induce action potentials that synchronize the cardiac tissue and therefore NCX is essential for cardiac development (please see for instance: Excitation-Contraction Coupling of the Mouse Embryonic Cardiomyocyte" Rapila R et al. 2008 and the other papers by the Tavi group or "Intracellular Ca^2^ Oscillations, a Potential Pacemaking Mechanism in Early Embryonic Heart Cells" Sasse P et al. 2007).*

*Furthermore, it is well known that intact Ca^2^ cycling is important for cardiac gene expression and heart development. The authors should clearly cite the previous work and specifically describe the novelty and mechanism of their study.*

We have now significantly expanded the Introduction to cite previous studies that have looked at early cardiac development and contractile function and more clearly highlighted the novelty of our data. The manuscripts mentioned in the comment above have looked at cells from later stages than us (linear heart tube stage or later) and moreover, have examined these cells in isolation from the embryo. We have done additional experiments with IP3 and RyR inhibitors and indeed, we have confirmed their findings of a later requirement of IP3 signalling (that serves as a positive control for our inhibitors), but have also made the novel finding that during the earlier stages that constitute the main focus of our manuscript, SR Ca^2^ is not required for the earliest manifestation of SACOs which precedes pacemaking activity.

*The main conclusion of the blocker experiments is that NCX reverse mode is important for Ca^2^ entry and gene regulation but how this should induce periodic beating or pacemaking has not been addressed or discussed. NCX and can only work in reverse mode when intracellular Na concentration is excessively high during an action potential (although controversial if this really happens, or if the effect is only a less effective forward mode). This has not been shown for embryonic cardiomyocytes and is highly unlikely because of the lack of Na channels or Na driven action potentials at this stage.*

We are focusing on the earliest onset of Ca^2^ influx, the establishment of Spontaneous Asynchronous Ca^2^ Oscillations (SACOs) prior to beating, and how this impacts on cardiomyocyte differentiation and crescent morphogenesis. The presence of SACOs precedes pacemaker activity and synchronisation which, are outside the focus and scope of this manuscript.

NCX has been shown to act in both forward and reverse mode in embryonic cardiomyocytes (Reppel et al. 2007, Annals of the New York Academy of Sciences) and the directionality of NCX1 activity is dependent on the concentration gradient of intra- and extra-cellular Ca^2^ and Na in addition to the membrane potential (Reppel, Reuter, et al. 2007). The conditions that favour reverse mode NCX1 activity include less depolarised membrane potentials. Reversal potential, for NCX1 is around -26mV. In terms of NCX1 function this means that, if the membrane potential was lower than -26mV, “forward mode” activity (Ca^2^ efflux) would occur, however at membrane potentials above -26mV “reverse-mode” activity (Ca^2^ influx) would predominate. Resting membrane potential of immature cardiomyocytes is known to be significantly higher than the adult (-85mV). Embryonic cardiomyocytes at around E10.5 have been shown to have a resting membrane potential of around -40mV (Sasse et al. 2007; Rapila et al. 2008), with less differentiated cardiomyocytes potentially having an even greater than -40mV membrane potential, increasing the likelihood and potential functional relevance of “reverse mode” NCX1 activity as well as making it more susceptible to changes in the concentration of intracellular Ca^2^ and Na. Thus NCX1 function within the stage 0 cardiac crescent could be responsible for both the Ca^2^ influx and efflux, correlating with the slow dynamics of stage 0 Ca^2^ transients, potentially regulating Ca^2^ homeostasis until contraction is initiated.

We have qualified our interpretation of NCX1 acting in reverse mode in the revised text:

“While the existence of the NCX reverse mode is contentious, the observation that at stage 0 SACOs were abolished with CB-DMB treatment but persisted when treated with nifedipine suggests that NCX and not the LTCC plays a major role in Ca^2^ influx at this early stage.”

*The (confusing) idea of a reverse mode NCX comes from the literature that state that at low dosages (EC50 in the low µM range) KB-R is preferential blocking a reverse mode. Although oocyte experiments with non-physiological intracellular Na concentrations suggest this, other experiments on cardiomyocytes (e.g. "Direction-independent block of bi-directional Na /Ca^2^ exchange current by KB-R7943 in guinea-pig cardiac myocytes" by Kimura, J et al. 1999) show that this specificity is not existent and other papers show that this is highly depending on intracellular Na concentrations and drug dosage used. Thus without electrophysiological analysis, it remains unclear, if the inhibitory effect on beating and cardiac crescent formation after block of NCX by KB-R or CB-DMB is due to effects on of Ca^2^ oscillations, pacemaking, translemmal Ca^2^ entry or exit, membrane potential fluctuation, synchronization by action potentials, imbalance of Ca^2^ homeostasis or off-target effects that occur especially when applied long term.*

We agree that the concept of NCX working in reverse mode is controversial despite the fact that it has been reported that KB-R9743 preferentially blocks reverse mode NCX1 activity (Hoyt et al. 1998; Iwamoto 2004). To this end we have revised the manuscript to reflect this (see response to comment above).

*In summary, the presented data, although in part novel and interesting, together with the known literature on embryonic cardiomyocytes and on the physiology of ECC, LTCC and NCX function does not allow to conclude a mechanism.*

We thank the reviewer for the positive comments on our data being novel and interesting. Given the extensive new data and our revised interpretation we believe we have now converged on a mechanism whereby NCX1 is required for the establishment of SACOs within the earliest crescent stages as a pre-requisite for cardiomyocyte differentiation and crescent morphogenesis.

*Specific major points:*

*The Introduction is too short and only very general. Specifically, several known facts on initiation of the first heart beat and the role of calcium gene regulation during embryonic heart development should be cited. For instance, several earlier publications have tried to address "the important question of when and how contractile activity of cardiomyocytes is first initiated during development" (see above). In addition to cite these earlier work in introduction it should be better pointed out, why these earlier studies are not sufficient and the current study is required (which is easy, because of the nice* in vivo *data).*

*Also the first paragraph of the section "A role for the Sodium-Calcium exchanger…" belongs to the Introduction. Please describe the well-known ECC in the adult heart a bit less but introduce more the known facts on ECC and the importance of NCX for pacemaking in the embryonic heart (see above).*

We appreciate the reviewer’s suggestions and hope that the revised manuscript now incorporates more thoroughly previous literature (Introduction) as well as clearly stating the novelty of the studies presented herein.

*Because of the low resolution of the supplied PDF file, the cardiac maturation and cross striation at the different crescent sages is difficult to review. Maybe providing inserts to highlight the specific patterns (especially stage 1 2) would help. Is cross striation completely absent in stage 0?*

We have now looked in more detail at cardiac maturation and sarcolemmal development within the developing cardiac crescent. In the revised manuscript we have now included analyses of two sarcomere related proteins (myomesin and sarcomeric α-actinin) as well as cardiac troponin T. In addition, we have performed qRT-PCR to assess the expression of these genes during crescent development. In both the revised Figure 1 and Figure 1—figure supplement 1, inserts are provided highlighting the formation of sarcomeres from stage 1 onwards. From this more detailed analysis we have been unable to detect cross striation within the stage 0 cardiac crescent, as is now addressed in the subsection “Staging of early cardiac development and sarcomeric assembly”.

*The finding of Ca^2^ transients at stage 0 is fascinating. Maybe better call them "spontaneous asynchronous Ca^2^ oscillations" also to highlight the difference semantically to the (action potential-based?) coupled transients at stage 1 . What is the average frequency and is there a correlation between frequency and TTP (maybe more interesting to show than TTP vs TT_1/2_M which lacks a conclusion)? Is there evidence that transients are coupled between individual cells (from the video this seems to be sometimes the case). Although not in the scope of the authors´ manuscript, recording of membrane potential is essential to understand the nature of cell-cell coupling of these fascinating data.*

We thank the reviewer for their positive comments and have taken on-board the suggested terminology of "spontaneous asynchronous Ca^2^ oscillations" (SACOs) to describe Ca^2^ “transients” at stage 0. We have now edited the figure containing characterisation of stage 0 transients and no longer include the data relating to TTP and TT_1/2_M. The revised Figure 2 now includes characterisation of SACOs within a 90 sec video and highlights the variation in frequency and oscillation dynamics. We believe there is evidence that transients are coupled between some individual cells and we are planning future studies to measure membrane potential within these cells. However, this requires extensive further experimentation which is beyond the scope of this submission, but will be important in understanding the progression of SACOs to synchronised/paced transients.

*Figure 3: Normalizing gene expression data to D0 or E7.5 (where most genes are not expressed) make less sense, adds noise and masks information. Please consider normalization to the time point of the highest value.*

After analysing the qRT-PCR data in multiple different ways we do not find an increase in noise when comparing at time point 0. Within the ESC model, there is variation in the extent of differentiation between cultures, which becomes greater the longer the differentiation time-course, therefore comparing to the latest stage (e.g. D14), as befits higher expression values, incurs a greater variation between experiments. This same applies to the longer-term embryo culture experiments.

*Unfortunately, Cacna1d (Cav1.3), which is the dominant isoform of LTCC in the early embryo (see "Subtype switching of L-Type Ca 2 channel from Cav1.3 to Cav1.2 in embryonic murine ventricle". Takemura H et al. 2005) was not analyzed and therefore the expression data of Cacna1c does not suggest a minor role of LTCC, but just highlights the well-known isoform switch (see also "Functional Embryonic Cardiomyocytes after Disruption of the L-type α1C (Cav1.2) Calcium Channel Gene in the Mouse"). The analysis of other LTCC genes or western blot at different stages from heart tissue might help in this regard.*

We thank the reviewer for this suggestion and have now investigated the expression of Cav1.3 (Figure 3—figure supplement 1; subsection “Ion channels are expressed during the earliest stages of heart development”). However, by performing pharmacological inhibitor studies using Nifedipine, we have effectively inhibited all LTCC isoforms and not just Cav1.2 (*Cacna1c*). The previous suggestion of a “minor role” for LTCC was based on the fact that the NCX inhibitor (CB-DMB) had a significant inhibitory effect on SACOs as compared to LTCC inhibition (Nifedipine) or control (DMSO). This is now further reinforced in the revised Figure 4 and we have revised the text to limit our conclusions, for relative roles of NCX versus LTCC, to the earliest manifestation of Ca^2^ handling.

*The establishment of an embryo culture for investigation of cardiac crescent or heart tube formation as well as for gene expression analysis is a definite plus and a main methodological advancement of this paper. Very nice! However, the fact that "Embryos cultured in CB-DMB were delayed in terms of cardiac crescent formation" is not supported by the one actin image shown (Figure 6O) although it is one of the main messages of the paper. Repeating of experiments, extensive histological analysis by specific crescent staining, and quantitation is suggested. Also a time course analysis of crescent formation with and without CB-DMB and KBR would enhance the paper. How many hours are the cardiac crescent forming delayed under NCX blockage?*

We thank the reviewer for their positive feedback on the embryo culture experiments. As suggested, we have further expanded these experiments on multiple cultured embryos per treatment (DMSO, n=6; CB-DMB, n=8; Nifedipine, n=6) and performed immunostaining for Nkx2.5 to mark cardiac progenitors within the forming crescent and cardiac troponin T to mark the more differentiated cardiomyocytes. These experiments revealed a significant decrease in the number of cTnT cells in CB-DMB-treated embryos in comparison to control or Nifedipine treatment (revised Figure 5) and clearly show the inhibitory effect of pharmacological NCX blockade on cardiomyocyte differentiation, which also correlated with a decrease in cardiac specific gene expression (revised Figure 5). Due to the clear and consistent decrease in cTnT staining after NCX inhibition, we have not quantified the immunofluoresence, particularly as quantification of immunostaining is of limited accuracy. However, we present quantitative data on changes in gene expression in the CB-DMB treated embryos through qRT-PCR. Whilst being able to study a developmental timecourse of cardiac crescent development with and without pharmacological inhibition would be ideal, it is not currently feasible due to issues associated with staging embryos, variations in extent of development between embryos and the technical limitations with the live-imaging.

*Was the data in Figure 5 obtained from Ca imaging or from video microscopy? If here video microscopy was used, also data from Ca imaging should be added to see the acute (>1-2 minutes) effects of the three blockers on Ca^2^ transients and remaining Ca^2^ oscillations (before all Ca^2^ has left the cell and the impaired Ca^2^ homeostasis is blocking pacemaking).*

Ca^2^ imaging was used with differential interference contrast (DIC) microscopy to obtain bright-field images. Imaging was conducted at 5 minutes after drug application to allow for diffusion and penetration of the embryo. Unfortunately, our current in vivo imaging setup does not allow us to image instantaneously after drug application, meaning we cannot currently observe acute changes in Ca^2^ dynamics which would be informative. Attempts to continuously image a single embryo impacted on viability and cardiac function due to excessive photo- damage. Instead we have focused on imaging multiple embryos (Figure 4 (stage1/2: DMSO, n=4; CB-DMB, n=7; KB-R7943, n=6; nifedipine, n=6; stage3/LHT: DMSO, n=5; CB-DMB, n=14; KB-R7943, n=7; nifedipine, n=8) at specific time- points (5mins, 15mins and 30mins).

*In the Discussion the authors have claimed to use "two independent pharmacological channel blockers CB-DMB and KB-R7943, in embryo cultures", but only show CB-DMB data in Figure 6. Please also show the KB-R data on embryonic cultures.*

We apologize for this confusion. While CB-DMB and KB-R7943 were used for the acute, short term embryo culture experiments, only CB-DMB and Nifedipine were used in the long term embryo culture studies. Due to the difficulties associated with obtaining large numbers of embryos at the same development stage (dissection, handling and maintaining constant culture conditions) and the fact that during later stages the inhibitors seem to have the same effect, we focused on CB-DMB to reduce the number of animals used for the experiments and to enable comparisons to be drawn with the parallel experiments conducted in ESC- cardiomyocyte cultures.

*The authors describe “sequential addition of CB-DMB to nifedipine-treated embryos after 15 mins, subsequently blocked the transients that were refractory to LTCC inhibition (not shown)." Please provide the data and statistics and explain the mechanism. What is the percentage of transients that were not blocked by LTCC inhibition? Was this observed at all stages or is this stage specific?*

These sequential inhibitor experiments were initially carried out, but have since been superseded by a more rigorous focus on the independent inhibitor studies, in light of the evident early versus later roles for NCX1 versus LTCCs. We have since edited the manuscript to remove the sequential inhibitor data.

[Editors’ note: the author responses to the re-review follow.]

*The manuscript resubmitted after revision is much improved and is now under review with a view to publication of these interesting findings. The authors show that the sodium calcium exchanger (NCX) early, at what they define as stage 0, is essential for generating spontaneous asynchronous Ca^2^ oscillations (SACO) in the cardiac crescent and that these are essential for cardiac gene expression and subsequent development. However, interpretation of the results and the proposal that reverse mode NCX, allowing Ca^2^ to enter the cell, is involved in the mechanism are still problematic.*

We thank the reviewers and editors for the assessment that the manuscript is much improved following the first round of revisions and for the progression towards review to publish our “interesting findings”. We have now toned down our interpretations on mechanism of reverse NCX1 function in the revised text as requested. In addition, we now include an additional set of KB-R7943 inhibitor experiments on stage 0 embryos and have re-normalised all of our real time qPCR gene expression data in line with the reviewer requests. Please find a detailed point-for-point response to the remaining issues detailed below.

*Firstly, in the experiments with inhibitors, the stage 0 experiments are critical. The authors use nifedipine to block voltage dependent Ca^2^ channels (VDCC), which interestingly at stage 0 has less effect on SACO, in contrast to later stages when it blocks activity. CB-DMB, which blocks NCX, affects SACO at stage 0, but it also blocks Ca^2^ exit mode. In the manuscript KB-R7943 is claimed to specifically block reverse mode NCX but it was not used at stage 0. This result should be shown. The finding that this inhibitor blocks activity at stages 1 and 2 is not so important because nifedipine does this too. In their interpretation the authors should be aware of controversy about the action of KB-R7943.*

We have now performed follow-up experiments using KB-R7943 at stage 0, and have included this data in the revised Figure 4. Whilst we are aware of the controversy surrounding the mode of KB-R7943 action (addressed in the fourth paragraph of the Discussion), we have found that two independent NCX inhibitors block the SACOs observed at Stage 0, allowing us to conclude that NCX is required for their generation.

*The SACO shown in Figure 2 and Video 3 is one of the most important pieces of data and in fact is the major novelty. Please provide more data here including n-numbers, statistics and report reproducibility of these findings. Are SACO observed in every stage 0 embryo? What is the Ca^2^ cycling frequency and amplitude and how often are single uncoupled (Figure 2´) and multiple coupled cells (Figure 2´´) observed?*

We have now provided more data to describe the stage 0 SACOs including the number of stage 0 embryos assessed (n=35) as well as more data on the Ca^2^ transient dynamics including the time to peak Ca^2^ signal (Ca^2^ influx) and the time to ½ maximum Ca^2^ signal (Ca^2^ efflux) (Figure 2—figure supplement 1). In general, as long as the endoderm was properly removed to allow penetration of the dye and better visualization of the cardiac mesoderm, SACOs were observed in every stage 0 embryo, revised text in subsection “The onset of Ca^2^ -handling in the cardiac crescent”. We believe that further studies on SACO dynamics, especially in terms of uncoupled and coupled cells, is beyond the scope of this current manuscript but is something we are actively addressing by developing methodology to accurately quantify Ca^2^ transient amplitude and synchronisation in live embryos, given this requires longer imaging periods than we are currently able to perform.

*The experiments on blocking SACO with CB-DMB in contrast to nifedipine lead to one of the most important findings, however the analysis and statistics are very limited. Please provide absolute numbers of cells with SACO and original data (example traces, videos) to enable the reader to follow the analysis and statistics. Line 310: What does " 0.87 0.083, mean SEM" refer to? How are cells without SACO identified if they do not show Ca^2^ signals? How is the variance of cells with SACO per embryo? ANOVA comparisons are required to take these variations in control embryos into account.*

Following on from the new experiments conducted with KB-R7943 at stage 0, we have reanalysed our data and now report SACO inhibition after drug treatment as the percentage inhibition of SACOs compared to baseline (subsection “Ca^2^ transients within the forming cardiac crescent are dependent upon NCX1”, last paragraph). We have also included a supplementary table of the original data analyses to document absolute number of SACOs observed before and after treatment, ratio of SACOs maintained after treatment, percentage inhibition, area containing SACOs and length of imaging (Figure 4, [Supplementary-material SD1-data]). Regarding the statistics, we have chosen to perform Fisher Exact Probability Test (FEPT) and treat this data as categorical, i.e. presence or absence of SACOs, since the variance for CB-DMB and KB-7943 is virtually non-existent (almost all embryos had zero SACOs after treatment) and hence statistical tests which assume a continuous data model are not suitable and cannot be applied to data.

*The novel Figure 3—figure supplement 1 on RyR/APD treatment is confusing as it does not follow the authors (very good) stage classification (stage 0, 1, 2, 3, LHT) but has a novel classification (turned/pre-turned embryo). Please label data as stage 0, stage 1 and LHT, at least. Please add statistics, error bars and original Ca^2^ imaging traces. A DMSO control for stage 3 is missing.*

We apologise that the later staging was confusing. This data was collected using embryos in which the heart had begun to loop, which is much later than the LHT stage and is why we cannot use our new staging method (applies to stages up to LHT). Stages later than the LHT represent a positive control for the Sarcoplasmic Reticulum inhibitor experiments, which we found had no effect at the earlier stage, (indicating the Ca^2^ at the earliest stages is not SR dependent). The term ‘turned’ is well accepted within the mouse embryology field and refers to a specific morphogenetic event when the entire embryo turns to assume a more conventional ‘foetal position’ within the yolk sac, curled up around its ventral aspect. The missing statistics, original Ca^2^ imaging traces and a DMSO control for stage 3 have now been included in a new Figure 4—figure supplement 1. The data lacks error bars because in this instance it is binary (either it was inhibited or not) hence also why we applied a Freeman-Halton extension of the Fisher exact probability test.

*Figure 4: the error bars are missing. In fact, it is not even fully clear from the methods what% inhibition means. Please discriminate between complete block of beating (movement in DIC images), reduction of frequency, complete block of Ca^2^ transients and reduction of Ca^2^ transient amplitude. Also please provide time courses and report the delay of drug action (should be easy as Ca^2^ imaging was performed at baseline and at 5, 15 and 30 minutes post drug treatment).*

*In general, in the text it is important to clarify what is meant by inhibition and partial inhibition.*

We have now included an explicit definition of what is meant by% inhibition in the Methods section (subsection “Live imaging – DIC, Ca2 561, Embryos and Cells”) and have removed the term partial inhibition as we appreciate this was misleading. The reason for no error bars is that our data is binary either – there were or were not Ca^2^ transients – and hence we have performed a Freeman-Halton extension of the Fisher exact probability test as the correct statistical analysis for this data type. We have also edited the text so that all inhibition data is based on Ca^2^ transient propagation as it is a clearer to assess the effect of pharmacological blockade. The Ca^2^ imaging data has now been reanalysed and we have revised Figure 4 to show the time-course of inhibition requested above. We are unable to determine Ca^2^ transient amplitude given the method for imaging Ca^2^ transients in intact tissue does not allow for ratiometric measurements.

*Although the pharmacological block using CB-DMB shown in Figure 4 lets the authors conclude that Ca^2^ entry though NCX is the basis for the SACO, the mechanism of how this could be periodic and lead to oscillations in Ca^2^ is unclear. The authors are correct that NCX can (under special circumstances such as elevated intracellular Na or depolarized membrane potential) lead to Ca^2^ entry, but once this occurs, the electrogenic nature of NCX (~3 Na for 1 Ca^2^ ) will again hyperpolarize the cell and export Na until the equilibrium potential is reached and Ca^2^ flux ceases. Thus it is not possible that the NCX can generate periodic Ca^2^ oscillations alone. Reverse mode only occurs if periodic action potentials are depolarizing the cell but then cells should be synchronized (no SACO). Maybe Na levels are cycling or the Na/K-ATPase is involved? This could be clarified by testing whether SACO at stage 0 are blocked by RyR/2-APB or by SERCA blockers (Thapsigargin/CPA). These mechanistic considerations should be taken into account in the authors' Discussion.*

We agree with the reviewer that the precise mechanism for periodic oscillations of Ca^2^ entry may involve other factors beyond NCX1, however, we have now shown, using two different inhibitors, that NCX is required for the generation of SACOs within the cardiac crescent at stage 0. We have taken into account the limitations on mechanistic insight as suggested and have addressed them within the revised Discussion, as well as detailing further experiments which would be of interest, but are currently beyond the scope of this manuscript.

*On the mechanistic front, as it stands the paper does not permit definitive conclusions; the authors must be less dogmatic about the SACO/Ca^2^ cycling mechanism and introduce notes of caution in their Discussion. The authors are encouraged to attempt electrophysiological analysis since without this it remains unclear if the inhibitory effect on beating and cardiac crescent formation of NCX by KB-R7945 or CB-DMB is due to effects on SACO, translemmal NCX reverse mode Ca^2^ entry (mechanism unclear) or NCX forward mode Ca^2^ exit (NCX block-based Ca^2^ overload can also inhibit gene expression), imbalance of Ca^2^ homeostasis or off-target effects. These analyses as well as straight forward siRNA experiments, proposed in the first reviews prior to resubmission, should be feasible with the elegant embryo culture technology set up by the authors.*

We have now reworded the text to be less dogmatic regarding the SACO/Ca^2^ cycling mechanism as well as introducing notes of caution within the revised Discussion. The siRNA experiments suggested by the reviewer, though elegant, are not straightforward. They involve optimisation of a delivery method (e.g. by electroporation, or virus), followed by prolonged culture to ensure knock-down of transcript and clearance of protein through turnover, only after which can the experiments be attempted. It is not clear that embryos subjected to such interventions will retain normal electrophysiological behaviour and what level of knock-down is required to ablate function. Therefore, whilst such experiments would be of interest and something we plan to pursue in the future we believe they are beyond the scope of this manuscript given the aims of the current study: to understand the effects of Ca^2^ handling and early contractile function on downstream cardiomyocyte differentiation and morphogenesis, during which we have provided strong evidence implicating a role for NCX1.

*A second aspect which requires further revision is the gene expression data. This should be normalised relative to housekeeping control genes. Normalising to E7.75 when most genes are not expressed and there is mainly noise is not informative, preventing meaningful comparison between genes. For instance, in the current presentation the time course of NCX and Cav1.3 seems to be superimposable. The data should be represented as dCT (normalised only to housekeeping genes) and then the meaningful differences in time course/relative levels discussed.*

We have now repeated all the analysis of qRT-PCR data and expressed everything as deltaCT as well as normalising only to housekeeping genes as suggested above by the reviewer(s).

[Editors’ note: the author responses to the re-review follow.]

*Before publication, the following remaining concerns of one reviewer will need to be addressed:*

*1) Please provide information on the frequency, duration and regularity of SACO at stage 0 as requested and as stated in the subsection “The onset of Ca^2^ -handling in the cardiac crescent” ("variable frequencies and durations"). From the authors comment in the rebuttal and from Video 4 (very nice) it is well understandable that precise numbers are difficult to give but at lease state something like "Compared to Ca^2^ transients at later stages (Figure 2) the SACO were rare. During a ~20 s recording period we observed only 10.3 -x individual SACO per embryo (n=24) occurring in different sites. Consecutive SACO in a same site were rarely observed within the 20 s imaging windows and therefore we conclude that SACO in individual cells occur at a frequency < 3 bpm."*

The suggested sentences above are now included with correct values on page 6:“Compared to Ca^2^ transients at later stages (Figure 2) SACOs were significantly slower, with fluorescence reaching peak intensity between 0.79 and 11.9 s and decreasing with a similar slow rate of efflux(Figure 2—figure supplement 1). During a ~20 s recording period we observed only 10.3 ± 0.7 individual SACOs per embryo (n=35) occurring in different sites. Consecutive SACOs in the same site were rarely observed within the 20 second imaging window and therefore we conclude that SACOs in individual cells occur at a frequency < 3 bpm.”

*2) Please describe the meaning of Figure 2—figure supplement 1 with something like "During a SACO, Ca^2^ rises slowly and reaches the peak between 0.5 and 10 s. Cells with slow Ca^2^ influx also showed similar slow Ca^2^ efflux (Figure 2—figure supplement 1)."*

We have incorporated an explanation similar to that suggested above about Figure 2—figure supplement 1 on page 6:

“SACOs were significantly slower, with fluorescence reaching peak intensity between 0.79 and 11.9 s and decreasing with a similar slow rate of efflux(Figure 2—figure supplement 1).”

*3) Because of the very slow nature of SACO, also oscillations in energy production and ATP levels can be involved (in fact the most likely explanation of SR is not involved). Please add this to the Discussion (fourth paragraph).*

We thank the reviewer for suggesting another potential mechanism of SACO generation and have now included this in the discussion on page 15:

“Due to the slow nature of SACOs, oscillations in energy production along with adenosine triphosphate levels could also be involved in SACO generation, especially with reduced SR function. Whilst SR inhibition did not prevent stage 3 Ca^2^ transients, we have not tested the involvement of SR function at stage 0 and, therefore, cannot fully exclude that periodic Ca^2^ releases from the SR are involved in the generation of SACOs. The latter has been reported in cultured E8.5-9 embryonic cardiomyocytes as a pacemaking mechanism (Sasse et al. 2007 and Rapilla et al. 2008), at later stages than were the focus in this study.”

*4) The authors state that Ryanodine 2-APB only affected embryos that had already undergone cardiac looping but the earliest stage tested was stage 3. Although requested, the effect of Ryanodine 2-APB on SACO was not analyzed. Thus it seems to be fair to include in the revised Discussion on the SACO mechanism (fourth paragraph) something like: "Because we have not tested the involvement of SR function at stage 0, we cannot fully exclude that periodic Ca^2^ releases from the SR are involved in the generation of SACO as reported before as a pacemaking mechanism in early embryonic cardiomyocytes (cite adequate references).*

We agree with the reviewer that we have not tested SR inhibition within the stage 0 embryo and have, therefore, included the following statement: **“**we have not tested the involvement of SR function at stage 0 and, therefore, cannot fully exclude that periodic Ca^2^ releases from the SR are involved in the generation of SACOs”.However, we respectfully disagree with inclusion of the “reported before” statement, since the two studies which have previously looked at the role of SR in pace making have studied cardiomyocytes cultured for 12 – 70 hours and collected at a significantly later time point (E8.5/E9) as stated in the Introduction. We, therefore, feel that referring to these studies directly against our early stage analyses is inappropriate; albeit our results recapitulate theirs when looking at the more mature hearts (i.e. looping at E8.5).